# Supportive consensus

**A. Palomares***, **M. Rebollo**, **C. Carrascosa**

VRAIn - Valencian Research Institute for Artificial Intelligence, Universitat Politècnica de València, Valencia, Spain

☯ These authors contributed equally to this work.
* apaloma@vrain.upv.es

**Data Availability Statement:** Data have been deposited to Figshare: https://doi.org/10.6084/m9.figshare.12625271.

**Funding:** The author(s) received specific funding for this work from the Valencian Research Institute for Artificial Intelligence (VRAIN) where the authors

## Abstract

The paper is concerned with the consensus problem in a multi-agent system such that each agent has boundary constraints. Classical Olfati-Saber's consensus algorithm converges to the same value of the consensus variable, and all the agents reach the same value. These algorithms find an *equality solution*. However, what happens when this *equality solution* is out of the range of some of the agents? In this case, this solution is not adequate for the proposed problem. In this paper, we propose a new kind of algorithms called *supportive consensus* where some agents of the network can compensate for the lack of capacity of other agents to reach the average value, and so obtain an acceptable solution for the proposed problem. Supportive consensus finds an *equity solution*. In the rest of the paper, we define the supportive consensus, analyze and demonstrate the network's capacity to compensate out of boundaries agents, propose different supportive consensus algorithms, and finally, provide some simulations to show the performance of the proposed algorithms.

## Introduction

This paper presents a novel approach to deal with consensus with bounded capacity nodes. To present our approach, let us suppose a straightforward example: three friends are going for dinner. When they receive the ticket, they find themselves in the following situation: it is 60 euros, and the first one only has 15 euros, the second one only has 20 euros, and the third one has 25 euros. According to the classic consensus algorithm of [1, 2], the only possible solution is an equality one; that is, each of them ought to pay 20 euros, but it is not possible since the first one only carries 15 euros, so there is no solution. Our approach is an equity one, so each one must give according to their available money, in this case, everything they have available (the first one 15 euros, the second one 20 euros, and the third one 25 euros).

In general, consensus problems consist of a group of entities that wants to reach an agreement about the value of a variable of interest in an incomplete information scenario.

### Scope of the paper

This paper defines a new kind of problems called supportive consensus (SC) to solve a state constraint consensus where agents must collaborate to find an equitable solution to the problem. SC allows to solve some situations that classical consensus model solutions are unable to

are currently working. This work is partially supported by the Spanish Government project RTI2018-095390-B-C31, GVA-CEICE project PROMETEO/2018/002, and TAILOR, a project funded by EU Horizon 2020 research and innovation programme under GA No 952215. The funders had no role in study design, data collection and analysis, decision to publish, or preparation of the manuscript.

**Competing interests:** The authors have declared that no competing interests exist.

do. For instance, when there is no overlapping between any two different agents' boundaries in the system. The only restrictions we impose for SC algorithms are the conservation of sum (a common restriction for consensus problems) and that all agents must converge to values within their boundaries. One significant result that we have proved is that if all the agents have their initial value inside their boundaries, equity-based solutions that the system could reach exist.

The paper also presents some proposals of algorithms that give a solution to the above-presented problem. These are not intended to be the best or the only possible solutions to the problem, but only possible ones. We want to emphasize that we have tried to make these algorithms 'as similar as possible' to the original consensus algorithm (Olfati-Saber). We present an exact algorithm that we have called SEA and a set of approximate algorithms (CORA, ICORA, RANA, and RACNA).

The approximate algorithms try to solve the problem by combining the consensus algorithm with a method that compensates values corresponding to agents that are 'out of its boundaries' in each iteration.

When these algorithms reach a stable state, a small amount of 'out of boundaries values' cannot be distributed in the network. For this reason, we have defined the relative error, which allows us to estimate the reliability of the solutions obtained.

Nevertheless, as they are the ones proposed till now, we have also included some simulation results using all these algorithms. These simulations are grouped into three sets according to the feature they want to test:

- First, a series of proofs of concept to follow in detail the proposed algorithms. With this purpose, a network of four nodes has been created. The mean value is outside the acceptable range of one of them, and the network must assume the excess. In this case, the algorithms that appear in Section "Supportive Consensus Algorithms Proposals" have been evaluated, and the evolution of the consensus value, the evolution of the sum of the excesses that the network must assume, and the evolution of the error–considered as the difference of the final value obtained concerning the average of the initial values–is shown.

- A second experiment studies how the error of the value reached concerning the mean of the initial values with each of the algorithms varies according to the network's size. This experiment shows how, as the network's size increases, the error is distributed throughout the network, and the deviation from the desired value is small.

- Finally, the third set of experiments evaluates the algorithms' behavior when the distribution of the intervals is unbalanced. We study what happens depending on the intervals' size, whether the solution is inside or outside most nodes, the distribution of nodes above or below the solution. The purpose is to show that the proposed solution is sufficiently robust, and the error obtained does not depend on the nodes' initial situation.

We want to highlight that 'supportive consensus' is an open problem, and that therefore other algorithms are possible.

## Related work

In general, the problems where *supportive consensus* could be applied are collaborative works, where all participating agents want to reach a consensus to fulfill a global goal. Moreover, some of the participants have bounded resources. Moreover, each agent participates according to its possibilities. This problem is very usual, not only in examples like the one commented above but in quite a lot of different domains, such as:

- Group Decision Making, where everyone will accept a solution only if it reflects the opinions of all of them, so this process requires some discussion and deliberation [3].

- Public good domain cooperation problems, where all people are interested in reaching the goal, but with heterogeneous contributing factors. An extreme case of this heterogeneity is the *free-rider problem*, where individuals do not contribute but want to share the global goal [4].

- *Cap and Trade*, or emissions trading, systems are government-regulated programs designed to limit, or cap, the total level of specific chemical by-products resulting from private business activity. Cap and trade's purpose is to create a market price for emissions or pollutants that did not previously exist and address possible negative externalities. The *Cap-and-Trade* system [5] uses the total *cap* to attain environmental goals and allows *trade* to achieve the effective scheduling through market regulation.

- Transmission rights markets, such as electric power or water rights markets [6], where there is a set of collaborating agents that want to share their rights to reach some goals. The establishment of tradable rights plays an essential role in improving the efficiency, equity, and sustainability of natural resource usage.

- Supply-demand balance in smart grids [7, 8], where a set of power stations must contribute to supply the global power demand of territory according to their boundaries.

- Safety or hazardous areas in the coordination of autonomous vehicles for rendezvous situations [9, 10], in which constraints on the position of vehicles are needed. In the same way, other [11, 12] present a consensus application to autonomous vehicle management with dynamic topologies.

As observed, they tend to be problems where there is a goal value to be reached, and all participating agents must contribute something to reach such goal: money, hours, energy, $CO_2$, or hours dedicated to a project by its participants. So, Knorn et al. [13] presents an interesting application of the classical consensus algorithm to be used by a fleet of hybrid electric vehicles to regulate $CO_2$ cooperatively.

A particular case dealt in the literature corresponds to a constrained consensus with common global constraints. So, in Shuai et al. [9], a global constraint is applied to the final consensus value. Agents reach a collective agreement value that must fulfill such a global constraint. So, they assume that all participating agents can reach the consensus. Furthermore, in Zhirong et al. [10], the global constraint is only known by some of the agents, and they all have the goal to achieve the minimum global aggregated cost. Similarly, [14] formalize the notion of scaled consensus wherein network components' scalar states reach assigned proportions, rather than a shared value in equilibrium.

Due to their relevant application to real-life problems, constrained consensus algorithms are a recent source of interest in the field. In real applications, there exist other constraints than system dynamics to be considered, such as communication bandwidth constraints [9], state constraints [9, 10], or velocity constraints [15] and input saturation [16]. These and other similar works related to constrained consensus processes on control dynamics can be found in the literature, such as second-order systems [17], and high dimensionality linear systems [18].

Fontan et al. and Hou et al. [19, 20] present similar approaches following the idea in classical consensus to agents with intervals, so they propose a modified classical consensus where all the intervals have to have a non-empty intersection where the solution will be contained, that is, an equality solution.

In contrast, we have called this new equity solution to a consensus *supportive consensus*. As commented, this solution is not directed by equality; that is, the mean value may not be the right solution. The approach presented in this paper can be classified as a state constraint consensus problem.

The rest of the paper presents some related work focusing on presenting the classical Consensus Algorithm, defining a *supportive consensus*. After that, some results are given about the system's capacity to calculate a problem's solution. Next, some proposals of algorithms are presented, followed by some simulation results. Lastly, some conclusions to the presented work are commented on.

## Background

The theoretical framework for solving consensus problems in dynamic agent networks was formally introduced by Olfati–Saber and Murray [1, 2]. This is one of the most promising research subjects in the Multi-Agent Systems (MAS) area that is currently emerging [21–26]. The agents' interaction topology is represented using edges of graphs, and *consensus* means reaching an agreement based on a certain amount that depends on the state of all agents in the network. This value represents the variable of interest in *agreement term* problem, which might be, for example, a physical quantity, a control parameter, or a price.

Let $G = \{V, E\}$ be an undirected, connected graph with $n$ nodes, $V = \{1, \ldots, n\}$, and $e$ edges $E \subseteq V \times V$, where $(i, j) \in E$ if there exists a link between nodes $i$ and $j$.

The graph can be represented by its weighted adjacency matrix $A = [a_{ij}]$. Let $(G, x)$ be the state of a network with value $x$ and topology $G$, where $x = (x_1, \ldots, x_n)^T \in \mathbb{R}^n$, and where $x_i$ is a real value that is associated with the node $i$. Á node's value might represent physical quantities measured in a distributed network of sensors (such as temperatures or voltages), or the amount of interest in a network of buyers and sellers in the market (prices, rights, or quality). A network is a dynamic system if $(G, x)$ evolves in discrete epochs. A consensus algorithm is an interaction rule that specifies the information exchange between agents and their neighbors to reach the agreement. The entire network reaches a consensus if and only if $x_i = x_j \forall i, j$. The distributed solutions of consensus problems in which no node is connected to all nodes are particularly interesting. The most commonly used consensus protocols are average, maximum, and minimum because they have broad applications in distributed decision-making multi-agent systems. It has been demonstrated [1, 2] that a convergent and distributed consensus algorithm in discrete-time (epochs) is:

$$x_i(t + 1) = x_i(t) + \varepsilon \sum_{j \in N_i} (x_j(t) - x_i(t)) \tag{1}$$

where $N_i$ denotes the set formed by all nodes connected to the node $i$ (neighbors of $i$). $0 < \varepsilon < 1/d_{max}$ is the step–size in the different iterations, where $d_{max}$ is the maximum degree in the network. In graph theory, the degree of a node $d_i$ is the number of edges that are incident to the node, and therefore $|N_i| = d_i$.

The algorithm converges to the average of the initial values of the state of each agent

$$\lim_{t \to \infty} x_i(t) = \frac{1}{n} \sum_{\forall i} x_i(0) = \bar{x} \tag{2}$$

and it allows the average for very large networks to be computed via local communication with their neighbors on a graph. Moreover, it has to be underlined that the sum of $x_i$ values is

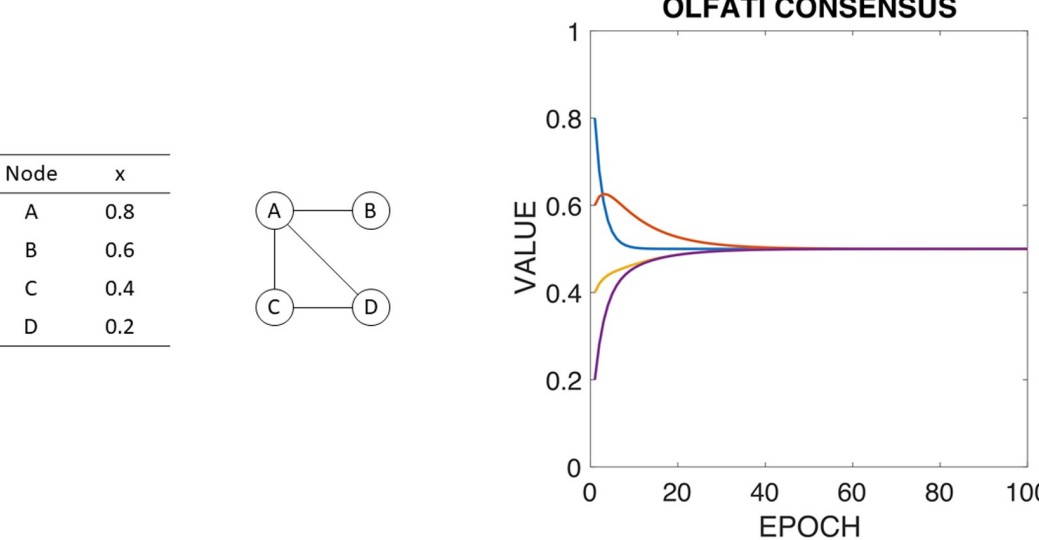

**Fig 1.** ABCD example: Graph (left), the initial values of nodes (center) and original Olfati-Saber's consensus algorithm for ABCD example without considering boundaries.

conserved along the process:

$$\sum_{\forall i} x_i(0) = \sum_{\forall i} x_i(t), \forall t \geq 0 \tag{3}$$

In case of weighted average consensus [27], Eq 4 is used to calculate the new values:

$$x_i(t + 1) = x_i(t) + \frac{\varepsilon}{w_i} \sum_{j \in N_i} (x_j(t) - x_i(t)) \tag{4}$$

where $w_i$ is the weight of agent i. In this case, the algorithm converges to the weighted average of the initial values.

To illustrate the working of the different consensus processes that will be defined in the paper, we will use a simple example formed by a graph with 4 nodes (A, B, C, and D) with the graph representation and initial values shown in Fig 1 left and center respectively. Fig 1 right shows the evolution of the Olfati-Saber's consensus (Eq 1) for this example.

If each node has defined a range of allowed values, four possible situations may happen:

- Stage 1: All ranges overlap, and the mean of the node's initial values is at the intersection of the overlapping of ranges. In this situation, the Olfati-Saber's consensus works as usually.

- Stage 2: There is one node with the mean value of the network's initial values outside of its range. Fig 2 presents a variation of the previous example where each node has defined a range of allowed values, and the mean value is outside of the range of allowed values of one of the nodes (D). Fig 2 left presents these ranges defined by their lower value ($x^{INF}$) and their upper value ($x^{SUP}$). Fig 2 center shows a graphical representation of the node ranges with their initial value inside them and how the average of these initial values (represented as an horizontal line in this graph) is outside node D range. Fig 2 right shows the evolution of Olfati-Saber's consensus for this situation, whereas node D reaches its upper limit of its range, he maintains such value, and cannot change it according to its neighbors' values. In this situation, this algorithm behaves in a *follow the leader* situation, where the the remaining nodes tend to the value of D-node's upper limit.

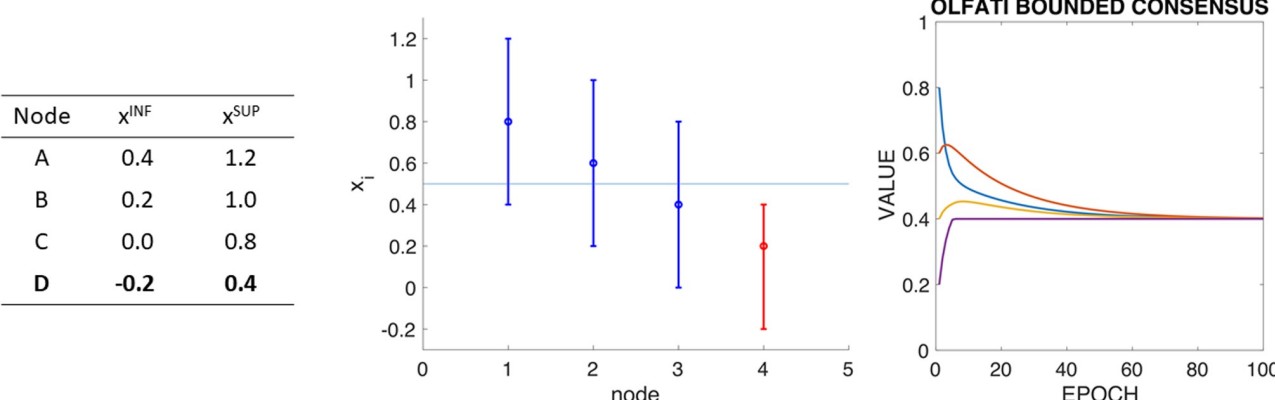

**Fig 2.** ABCD example with boundaries (I): Boundaries of each node (left), graph representation of boundaries and initial values (center), and Olfati-Saber's consensus algorithm. It has to be taken into account that the mean value of the network is 0.5, and this value is out of the node D boundaries.

As an agent is reaching its limit, it would remain in this value and attract the rest of the agents to converge. So, they reach a consensus, but the total sum of the values is not conserved. The initial total sum is two, and the final total sum is 1.6.

- Stage 3: There is more than one node with the mean value of the network's initial values outside their ranges. Fig 3 presents a variation of the previous example where each node has defined a range of allowed values. The mean value is outside of the range of allowed values of all the nodes, and there are no two nodes with overlapping ranges. This situation is extreme for this kind of problem. Fig 3 left presents the ranges of the nodes in the system, while Fig 3 center presents a graphical representation of the ranges and initial values of the nodes, with the mean of the initial values as a horizontal line. Fig 3 right presents the evolution of Olfati-Saber's consensus in this situation. In this case, the sum of the final values is 2.18, while the sum of the initial values is 2.

So, in general, Olfati-Saber's algorithm does not deal appropriately with constrained agents.

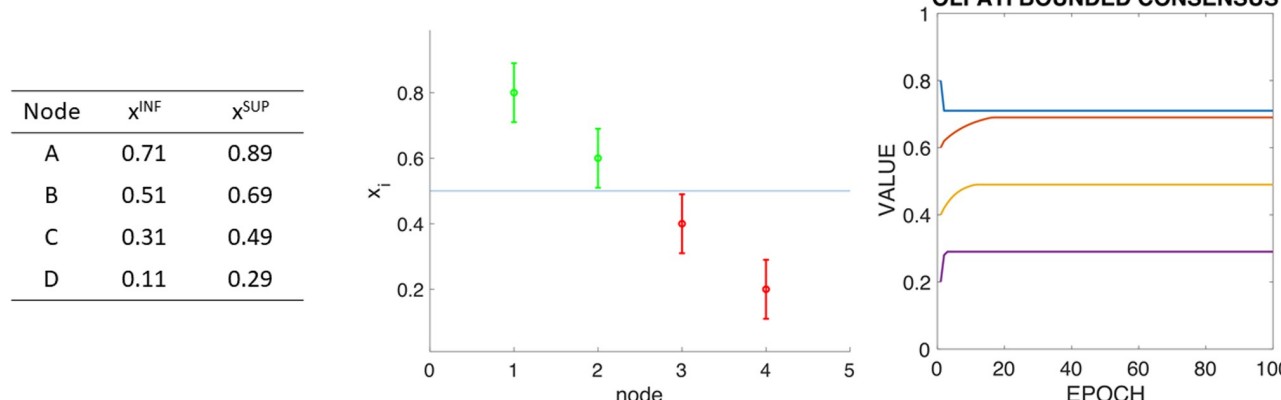

**Fig 3.** ABCD example with boundaries (II): Boundaries of each node (left), graph representation of boundaries and initial values (center), and Olfati's consensus algorithm. It has to be taken into account that no node boundaries overlap with no other. In this case, the sum of the final values is 2.18, while the sum of the initial values is 2.

## Supportive consensus

### Definition

Let us suppose that the nodes in the consensus process have a constrained range of possible values for the variable $x_i$ between its lower ($x_i^{INF}$) and upper ($x_i^{SUP}$) limits:

$$x_i(t) \in [x_i^{INF}, x_i^{SUP}] \tag{5}$$

Let us call $r_i = x_i^{SUP} - x_i^{INF}$ the *range* of the node i.

According to this new feature of the nodes, Olfati consensus model can be reached only if the intersection of all intervals is not empty and, therefore, the mean value belongs to this intersection:

$$\bar{x} \in \bigcap_{\forall i} [x_i^{INF}, x_i^{SUP}] \tag{6}$$

Let us define a *supportive consensus* as a state-constrained, first-order, consensus process where there exist constrained ranges for the nodes. Some nodes may have the mean value out of their bounds, but the nodes inside the bounds assume the corresponding part of the nodes in their limits so that the mean value is the same as if the whole nodes were unbounded.

The general supportive consensus has to fulfill Eq 3 as general classical consensus does.

Let $(G, x)$ be an undirected and strongly connected graph and V the corresponding set of nodes, as defined in Background Section. We represent a partition of the V set as

$$V = V^-(t) \cup V^*(t) \cup V^+(t) \tag{7}$$

where

- $V^-(t) = \{i | x_i(t) < x_i^{INF}\}$, is the set of nodes which their lower bounds are over the consensus value at epoch t.

- $V^*(t) = \{i | x_i^{INF} \leq x_i(t) \leq x_i^{SUP}\}$, is the set of nodes which intervals include the consensus value at epoch t.

- $V^+(t) = \{i | x_i(t) > x_i^{SUP}\}$, is the set of nodes which their upper bounds are under the consensus value at epoch t.

We define the excess of node $i \in V^+(t)$ at epoch $t$ that must be compensated *supportively* by the rest of the system $\delta_i^+(t)$ as

$$\delta_i^+(t) = x_i(t) - x_i^{SUP} \tag{8}$$

By definition $\delta_i^+(t) \geq 0$.

Symmetrically, the defect of node $i \in V^-(t)$ at epoch $t$ that must be compensated *supportively* by the rest of the system $\delta_i^-(t)$ as

$$\delta_i^-(t) = x_i(t) - x_i^{INF} \tag{9}$$

By definition $\delta_i^-(t) \leq 0$.

### Notation

A summation criteria notation will be used in the rest of the paper to improve its readability (capital letters without indexes mean that the sum has been calculated).

- $X = \Sigma_i x_i(t)$ is the sum of the current values of V.

- $X_{INF}$ is the sum of the lower values of $V$.

- $X_{SUP}$ is the sum of the upper values of $V$.

- $X^-(t)$ is the sum of the current values of $V^-(t)$.

- $X_{INF}^-(t)$ is the sum of the current lower bounds values of $V^-(t)$.

- $X_{SUP}^-(t)$ is the sum of the current upper bounds values of $V^-(t)$.

- $R^-(t)$ is the sum of the ranges values of $V^-(t)$.

- $X^*(t)$ is the sum of the current values of $V^*(t)$.

- $X_{INF}^*(t)$ is the sum of the current lower bounds values of $V^*(t)$.

- $X_{SUP}^*(t)$ is the sum of the current upper bounds values of $V^*(t)$.

- $R_{SUP}^*(t) = X_{SUP}^*(t) - X*(t)$.

- $R_{INF}^*(t) = X*(t) - X_{INF}^*(t)$.

- $R^*(t)$ is the sum of the ranges values of $V^*(t)$. $R*(t) = R_{SUP}^*(t) + R_{INF}^*(t)$.

- $X^+(t)$ is the sum of the current values of $V^+(t)$.

- $X_{INF}^+(t)$ is the sum of the current lower bounds values of $V^+(t)$.

- $X_{SUP}^+(t)$ is the sum of the current upper bounds values of $V^+(t)$.

- $R^+(t)$ is the sum of the ranges values of $V^+(t)$.

- $\Delta^-(t)$ is the sum of the defect values $\delta_i^-(t)$ of all nodes $i \in V^-(t)$ that must be compensated supportively. Therefore $\Delta^-(t) = X^-(t) - X_{INF}^-(t)$.

- $\Delta^+(t)$ is the sum of the excess values $\delta_i^+(t)$ of all nodes $i \in V^+(t)$ that must be compensated supportively. Therefore $\Delta^+(t) = X^+(t) - X_{SUP}^+(t)$.

By definition $\Delta^-(t) \leq 0$ and $\Delta^+(t) \geq 0$.

The total amount of out of boundaries values that have to be compensated by a supportive consensus algorithm is:

$$\Delta(t) = \Delta^-(t) + \Delta^+(t) = X^-(t) - X_{INF}^-(t) + X^+(t) - X_{SUP}^+(t) \tag{10}$$

Fig 4 summarizes graphically these concepts.

## Analysis of supportive capacity

In general, the values of the nodes are all inside their corresponding boundaries (Eq 5) and therefore

$$X_{INF} \leq X \leq X_{SUP} \tag{11}$$

Considering the partition of $V$ in their corresponding subsets (Eq 7):

$$X_{INF} = X_{INF}^-(t) + X_{INF}^*(t) + X_{INF}^+(t) \tag{12}$$

and

$$X = X^*(t) + X^-(t) + X^+(t) \tag{13}$$

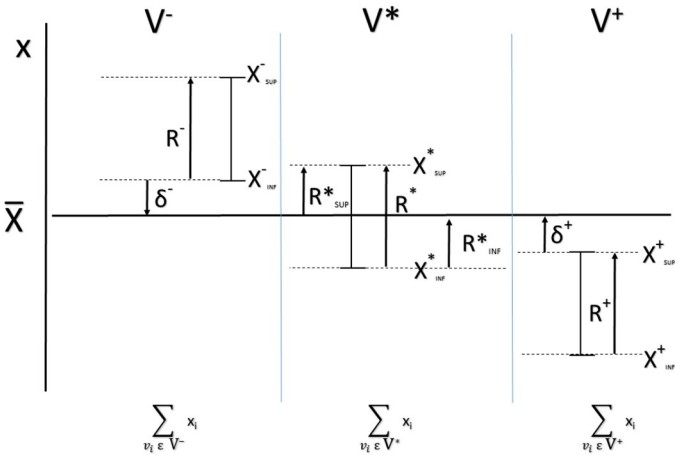

**Fig 4. Graphical representation of the node sets classification according to its bounds and accumulated magnitudes (temporal dependence not included for clarity).**

and

$$X_{SUP} = X_{SUP}^-(t) + X_{SUP}^*(t) + X_{SUP}^+(t) \tag{14}$$

and replacing in Eq 11, for the left inequality:

$$X_{INF}^-(t) + X_{INF}^*(t) + X_{INF}^+(t) \leq X^*(t) + X^-(t) + X^+(t) \tag{15}$$

and for the right inequality:

$$X^*(t) + X^-(t) + X^+(t) \leq X_{SUP}^-(t) + X_{SUP}^*(t) + X_{SUP}^+(t) \tag{16}$$

Taking into account Eq 10, for the left inequality:

$$X_{INF}^-(t) + X_{INF}^*(t) + X_{INF}^+(t) \leq X^*(t) + X_{INF}^-(t) + X_{SUP}^+(t) + \Delta(t) \tag{17}$$

and for the right inequality

$$X^*(t) + X_{INF}^-(t) + X_{SUP}^+(t) + \Delta(t) \leq X_{SUP}^-(t) + X_{SUP}^*(t) + X_{SUP}^+(t) \tag{18}$$

Let's analyze each one of the terms of this inequality separately. With the left inequality

$$
\begin{aligned}
X_{INF} - X_{INF}^-(t) &\leq X^*(t) + X_{SUP}^+(t) + \Delta(t) \\
X_{INF}^+(t) + X_{INF}^*(t) &\leq X^*(t) + X_{SUP}^+(t) + \Delta(t) \\
X_{INF}^*(t) - X^*(t) + X_{INF}^+(t) - X_{SUP}^+(t) &\leq \Delta(t) \\
-\underbrace{(X^*(t) - X_{INF}^*(t))}_{R_{INF}^*(t)} - \underbrace{(X_{SUP}^+(t) - X_{INF}^+(t))}_{R^+(t)} &\leq \Delta(t)
\end{aligned}
$$

$R_{INF}^*(t)$ is the capacity to compensate differences of other nodes by the ones which values are inside their boundaries, and $R^+(t)$ is the whole capacity defined by the boundaries of the nodes which values are above their upper limits (see Fig 4). We define the network capacity to assume values under the lower limits of the nodes as

$$C_{INF}(t) = R_{INF}^*(t) + R^+(t). \tag{19}$$

By definition $C_{INF}(t) \geq 0$. Let us analyze the right inequality

$$
\begin{aligned}
X^*(t) + X_{INF}^-(t) + \Delta(t) &\leq X_{SUP} - X_{SUP}^+(t) \\
X^*(t) + X_{INF}^-(t) + \Delta(t) &\leq X_{SUP}^*(t) + X_{SUP}^-(t) \\
\Delta(t) &\leq \underbrace{X_{SUP}^*(t) - X^*(t)}_{R_{SUP}^*(t)} + \underbrace{X_{SUP}^-(t) - X_{INF}^-(t)}_{R^-(t)}
\end{aligned}
$$

$R_{SUP}^*(t)$ is the capacity to compensate excesses of other nodes by the ones which values are inside their boundaries, and $R^-(t)$ is the whole capacity defined by the boundaries of the nodes which values are under their lower limits (see Fig 4).

We define the network capacity to assume values over the upper limits of the nodes as

$$
C_{SUP}(t) = R_{SUP}^*(t) + R^-(t). \tag{20}
$$

By definition $C_{SUP}(t) \geq 0$. Combining both inequalities, we get

$$
-C_{INF}(t) \leq \Delta(t) \leq C_{SUP}(t) \tag{21}
$$

In general, assuming that the initial values of all agents ($t = 0$) are within their bounds, the network globally always ($\forall t$) has enough capacity to compensate the total amount of values $\Delta(t)$ corresponding to the agents for whom the mean (Olfati solution) is outside its boundaries. That is, the total amount of out of boundaries values $\Delta(t)$ can be compensated by the rest of the network whenever the agent's initial values are between their bounds.

The supportive consensus is a new kind of problem, in contrast to classical consensus, where the solution must be satisfied in any case, regardless of all network nodes' boundaries.

Once that has been demonstrated that supportive solutions can exist, the rest of the paper gives some proposals of possible supportive consensus algorithms and some simulation experiments over synthetic networks.

## Supportive consensus algorithms proposals

The supportive consensus algorithms that we introduce in this paper are similar to Olfati's because they must keep the mean (restriction), but also they must allow that some agents in the network can compensate the $\Delta(t)$, and thus obtaining satisfactory agreements for all of them.

The algorithms that we have called "Supportive Consensus Generic Algorithms" try to solve the problem by combining the consensus algorithm with a method that allows the delta to be distributed in each iteration. The algorithm that we have called CORA simply performs a consensus iteration to distribute the agent's deltas. The algorithm that we have called ICORA performs n consensus iterations to distribute the delta. The algorithms that we have called RANA and RACNA are variations of the CORA algorithm, which try to optimize the distribution of the deltas, taking into account the neighboring node's characteristics.

The algorithm that we have called SEA provides an exact solution to the problem of SC, although the consensus is made on a transformed variable, not on the original one.

## Supportive exact solution

It is possible to determine the exact solution $x_e$ for the SC process in a scenario with perfect information. In such a scenario, we have information from the complete network, and it is possible to calculate the consensus solution with a centralized procedure. We consider the exact solution as the solution in which all the nodes move towards the mean value, staying at

their limits if they cannot reach the average value. The rest of the nodes will share the same value, compensating for the difference.

The process to calculate the exact solution in a centralized way follows Algorithm 1. This version considers for clarity that the solution is over the upper bounds.

**Algorithm 1** Exact solution in a scenario with perfect information

```
1: x_e = X/n
2: x_i = x_e ∀i ∈ V
3: V* = V
4: while ∃i | x_i > x_i^SUP do
5:     δ_i = (x_i − x_i^SUP)/|V*|
6:     x_i = x_i^SUP
7:     V* = V* − {i}
8:     V^+ = V^+ + {i}
9:     x_j = x_j + δ_i ∀j ∈ V*
```

Let us consider ABCD example, with $x = \{8, 6, 4, 2\}$ and intervals $R = \{[5, 10], [4, 6], [2, 5.2], [0, 4]\}$. In this situation., $x_e = 5$, which is out of D's bounds. Therefore, $D$ cannot reach the consensus value and the remaining nodes compensate the excess. $V^* = \{A, B, C\}$ and $V^+ = \{D\}$, being $\delta_D = \frac{5-4}{3} = 1/3$. Nevertheless, C cannot assume all this excess and the rest (1/3—0.2) must be compensated by the two other nodes. Finally, $V^* = \{A, B\}$ and $V^+ = \{C, D\}$, with $x_D = 4$, $x_C = 5.2$, $x_A = x_B = 5 + 1/3 + 0.2/3 = 5.4$

It is important to notice that Eq 21 ensures that, even arriving at the extreme case with one remaining node, $|V^*| = 1$, the node always can compensate the final pending amount.

## Supportive consensus generic algorithm

The proposed algorithms use a double layer network, in which the first layer stores the values of the different nodes $x_i(t)$, and the second layer stores the deviations of such values out of their bounds $\delta_i(t)$.

When a node $k$ over/undertakes its limit $x_k^{SUP}(x_k^{INF})$, the remaining passes to the other layer, that performs the Supportive Delivery function over $\delta_i$. After that phase, the *over/underrange* residual is included back again in the variable, and the process begins anew. The effect is that the *over/underrange* remaining is spread and diluted over the complete set of nodes.

These algorithms are based on the conservation of the sum of the initial values and follow the generic structure that can be seen in the algorithm 2:

**Algorithm 2** Supportive consensus generic algorithm.

```
1: δ_i'(t) = Supportive_delivery(i, t)
2: z_i(t) = x_i(t) + δ_i'(t)
3: z_i'(t) = z_i(t) + ε∑_{j∈N_i}[z_j(t) − z_i(t)]
```
$$4: (x_i(t+1), \delta_i(t+1)) = \begin{cases} (z_i'(t), 0) & \text{if } z_i'(t) \in V^*(t) \\ (x_i^{SUP}, z_i'(t) - x_i^{SUP}) & \text{if } z_i'(t) \in V^+(t) \\ (x_i^{INF}, z_i'(t) - x_i^{INF}) & \text{if } z_i'(t) \in V^-(t) \end{cases}$$

This algorithm begins calculating the delivery of the out of range values of the nodes (Algorithm 2, line 1). The only difference between the different proposed algorithms is how they define this *Supportive_delivery*(i, t) function. Then Algorithm 2, line 2 updates each node value, adding the delivered residuals. After that, Algorithm 2, line 3 performs a basic consensus iteration. Finally, Algorithm 2, line 4 classifies the nodes in their corresponding sets ($V^-(t)$ or $V^*(t)$ or $V^+(t)$) and calculates the values $x_i(t + 1)$ and residuals $delta_i(t + 1)$ for the next iteration of the algorithm.

The generic algorithm must conserve the sum of the initial values. Let us suppose that the Generic Algorithm 2, line 1 conserves the sum:

$$\Delta'(t) = \sum_{\forall i} Supportive\_delivery(i, t) = \Delta(t) \tag{22}$$

Taking into account that in the Algorithm 2, the accumulated values can be separated into their corresponding sets (line 2), line 3 performs a basic consensus that conserves the sum (Eq 3), and line 4 classifies the nodes in their corresponding sets, then:

$$
\begin{aligned}
X(t) + \Delta(t) \quad &= X(t) + \Delta'(t) = Z(t) = Z'(t) = Z'^-(t) + Z'^*(t) + Z'^+(t) \\
&= Z'^-(t) + Z'^*(t) + Z'^+(t) + X_{INF}^-(t) - X_{INF}^-(t) + X_{SUP}^+(t) - X_{SUP}^+(t) \\
&= X_{INF}^-(t) + (Z'^-(t) - X_{INF}^-(t)) + Z'^*(t) + X_{SUP}^+(t) + (Z'^+(t) - X_{SUP}^+(t)) \\
&= X^-(t+1) + \Delta^-(t+1) + X^*(t+1) + X^+(t+1) + \Delta^+(t+1) \\
&= X(t+1) + \Delta(t+1)
\end{aligned}
$$

Summarizing,

$$X(t) + \Delta(t) = X(t+1) + \Delta(t+1) \tag{23}$$

The convergence of the Olfati–Saber algorithm is guaranteed if the graph forms one connected component and $\varepsilon < \frac{1}{\max d_i}$. As Algorithm 2 conserves the sum of the initial values and fulfill the conditions of the Olfati–Saber consensus algorithm to converge, we need to ensure that the *Supportive_Consensus()* function also conserves the sum. We check this condition in each one of the proposed algorithms.

## Algorithms proposed

The algorithms that we propose in this section follow the Supportive Consensus Generic Algorithm 2 exposed in the last section. These algorithms try to deliver the out of range values of the agents. The only difference between them is how they define the Supportive Delivery Function. This Supportive Delivery Function must conserve the sum (Eq 22) for each of them. The proposed algorithms are:

1. Consensus Over Residuals Algorithm (CORA).

2. Iterated Consensus Over Residuals Algorithm (iCORA).

3. Residuals Among Neighbors Algorithm (RANA).

4. Residuals Among Capable Neighbors Algorithm (RACNA).

It is also important to note that these algorithms do not allow us to obtain exact solutions to the proposed problem. These algorithms tend to converge towards non-zero solutions of the $\Delta(t)$, usually with $\Delta(t) << X(t)$. $\Delta(t)$ depends on the algorithm and also on other factors, such as the structure of the network, the agent's initial values, or the ranges of variation of the agents. For this reason, in order to show and compare the performance of these algorithms we use the relative error ($re(t)$) defined as:

$$re(t) = \frac{\Delta(t)}{X(t) + \Delta(t)} \tag{24}$$

Next, we are going to show the different algorithms proposals.

**Consensus Over Residuals Algorithm (CORA).** This method consists of making one basic consensus iteration between all agents to distribute among all of them this *over/under-range*. Formally, this process is modeled by the following equations for each node:

$$Supportive\_delivery(i, t) = \delta_i(t) + \varepsilon \sum_{j \in N_i} [\delta_j(t) - \delta_i(t)] \tag{25}$$

*Sum conservation.* The CORA algorithm conserves the sum because this Supportive Delivery function is one iteration of the Olfati-Saber's consensus algorithm in the deviations layer that conserves their sum (Eq 3).

Fig 5 shows the result of the *CORA* for the supportive consensus in the ABCD example.

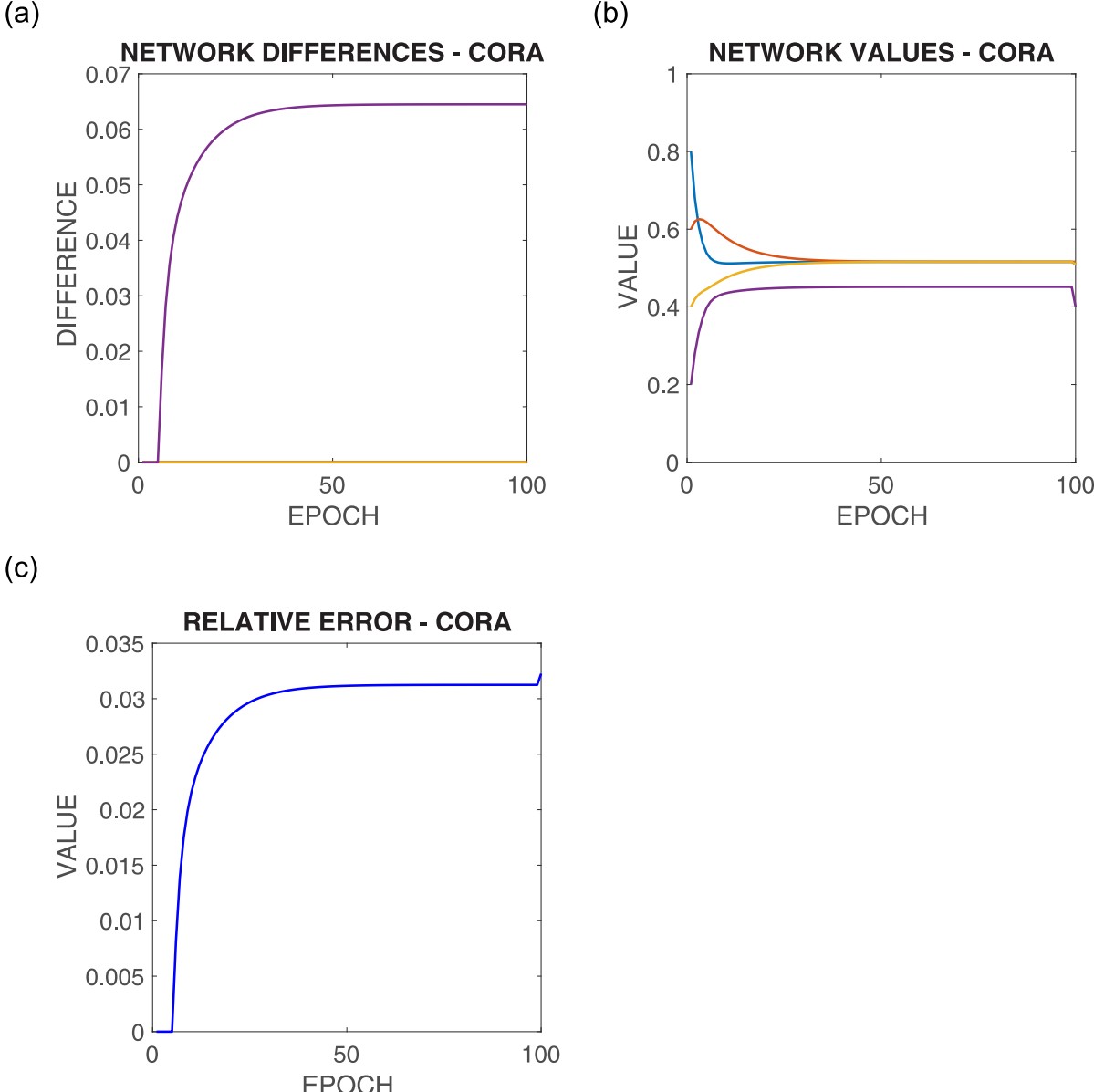

**Fig 5. Supportive consensus process according to the CORA algorithm in the ABCD example.**

So, Fig 5 left shows the evolution of $\delta_i(t)$ in the ABCD example (in this case, only $\delta_D(t)$ is not 0). Fig 5 center shows the evolution of $x_i(t)$ values, and Fig 5 right presents the evolution of $re(t)$.

The main drawback of this method is that making a consensus to deliver the $\Delta(t)$ between all the nodes, would make that, for instance, a node $k$ that have reached its upper bound, and its value is now $x_k^{SUP}$ could receive a $\delta_k(t) > 0$, and so it would end the algorithm with a value out of its bounds.

**Iterated Consensus Over Residuals Algorithm (iCORA).**   This algorithm is an extension of the CORA algorithm that consists of making $n \geq 1$ basic consensus iterations of the $\delta_i(t)$ layer between all agents to distribute among all of them this *over/underrange*.

Formally, the *Supportive_delivery*$(i, t)$ is calculated by the Algorithm 3.

**Algorithm 3** Iterative Consensus Over Residuals Algorithm (iCORA).
1: $\delta_i'(0) = \delta_i(t)$
2: **repeat**
3:     $\delta_i'(s + 1) = \delta_i'(s) + \varepsilon \sum_{j \in N_i} [\delta_j'(s) - \delta_i'(s)]$
4: **until** n iterations
5 **return** $\delta_i'(n)$

*Sum conservation*. The iCORA algorithm conserves the sum because the Supportive Delivery function is composed of n iterations of the Olfati-Saber's consensus algorithm in the deviations layer that conserves their sum (Eq 3).

Fig 6 shows the result of the *iCORA* for the supportive consensus in the ABCD example.

Fig 6 left shows the evolution of $\delta_i(t)$ in the ABCD example (in this case, only $\delta_D(t)$ is not 0). Fig 6 center shows the evolution of $x_i(t)$ values, and Fig 6 right presents the evolution of $re(t)$.

**Residuals Among the Neighbors Algorithm (RANA).**   According to this method, each node $i$ distributes its *over/underrange* $\delta_i(t)$ equally among each one of its neighbors ($\forall j \in N_i$). Therefore:

$$\delta_{i \to j}(t) = \frac{\delta_i(t)}{|N_i|} \forall j \in N_i \to \delta_i(t) = \sum_{j \in N_i} \delta_{i \to j}(t) \tag{26}$$

On the other hand each node $i$ receives the corresponding *over/underrange* $\delta_{j \to i}'(t)$ distributed by each one of its neighbors ($\forall j \in N_i$). Therefore:

$$\delta_{j \to i}'(t) = \frac{\delta_j'(t)}{|N_j|} \forall j \in N_i \to \delta_i'(t) = \sum_{j \in N_i} \delta_{j \to i}'(t) \tag{27}$$

In this case, the function that calculates the new $\delta_i(t)$ for each node $i$ according to the different delta values arriving to the node $i$ from its neighbors will be:

$$Supportive\_delivery(i, t) = \sum_{j \in N_i} \frac{\delta_j'(t)}{|N_j|} \tag{28}$$

*Sum conservation*. The RANA algorithm conserves the sum because this Supportive Delivery function conserves the sum. The reason is that this function only exchanges *over/underrange* values between the nodes in the layer of deviations, and therefore the sum is conserved.

Fig 7 shows the execution of the RANA algorithm in the *ABCD* example. So, Fig 7 left shows the evolution of $\delta_i(t)$ in the ABCD example. Fig 7 center shows the evolution of $x_i(t)$ values, and Fig 7 right presents the evolution of $re(t)$.

**Residuals Among the Capable Neighbors Algorithm (RACNA).**   The process uses the same approach as *CORA* does. It considers two layers for the network: one for the consensus

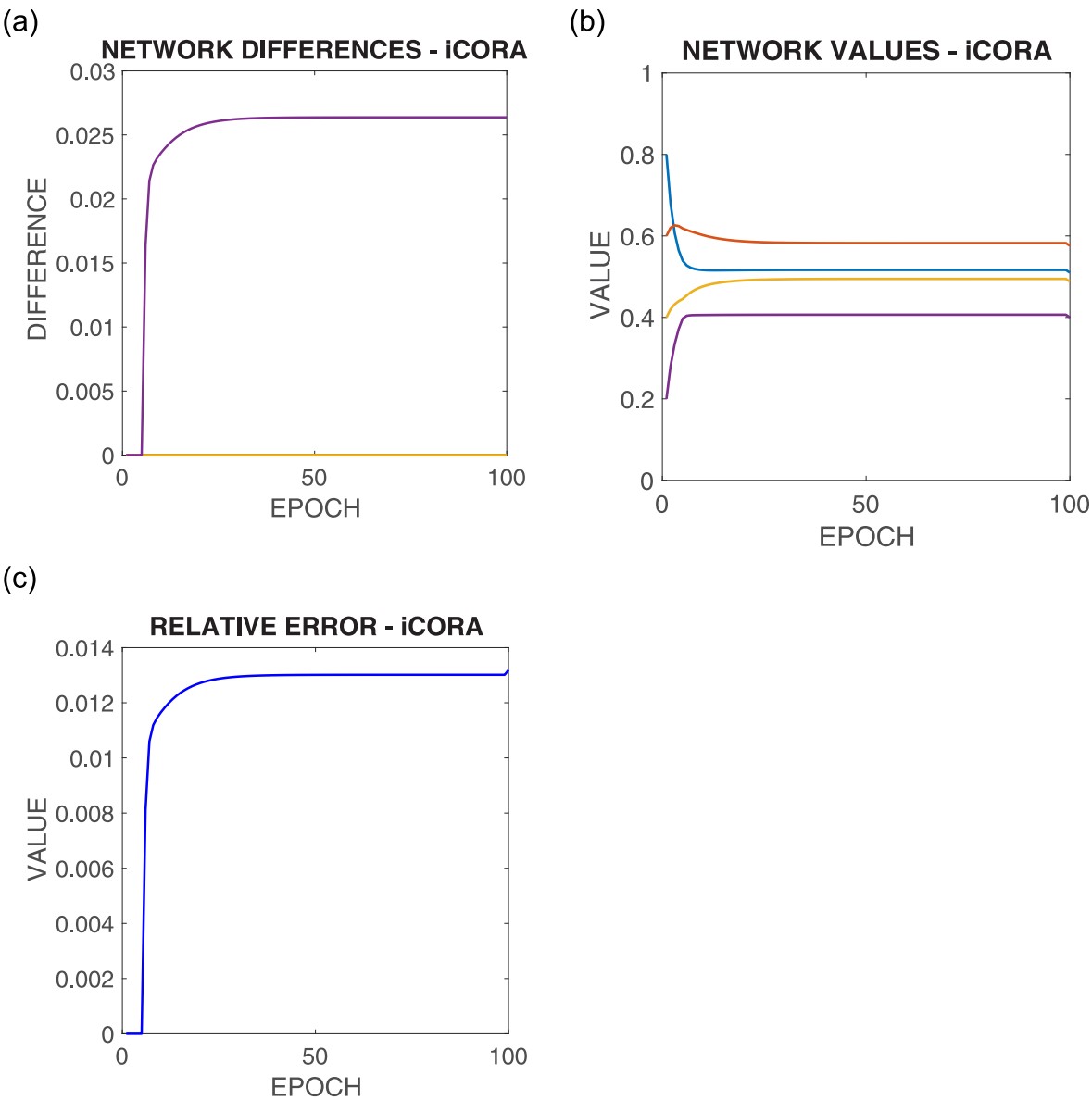

**Fig 6. Supportive consensus process according to the iCORA algorithm in the ABCD example.**

over the variable of interest, and a second one in which the $\delta_i(t)$ values are negotiated. In this case, the *over/underrange* $\delta_k(t)$ is proportionally delivered only among the capable neighbors. In this case, the function to deliver the *over/underrange* $\delta_i(t)$ is defined as indicated in Eq 29.

$$Supportive\_delivery(i, t) = \sum_{j \in N_i} \frac{\delta_j(t)}{|N_j \cap V^*(t)|} \tag{29}$$

*Sum conservation.* The RACNA algorithm conserves the sum because this Supportive Delivery function conserves the sum. The reason is that this function only exchanges *over/underrange* values between the nodes in the layer of deviations, and therefore the sum is conserved.

Fig 8 shows the execution of the RACNA algorithm in the *ABCD* example.

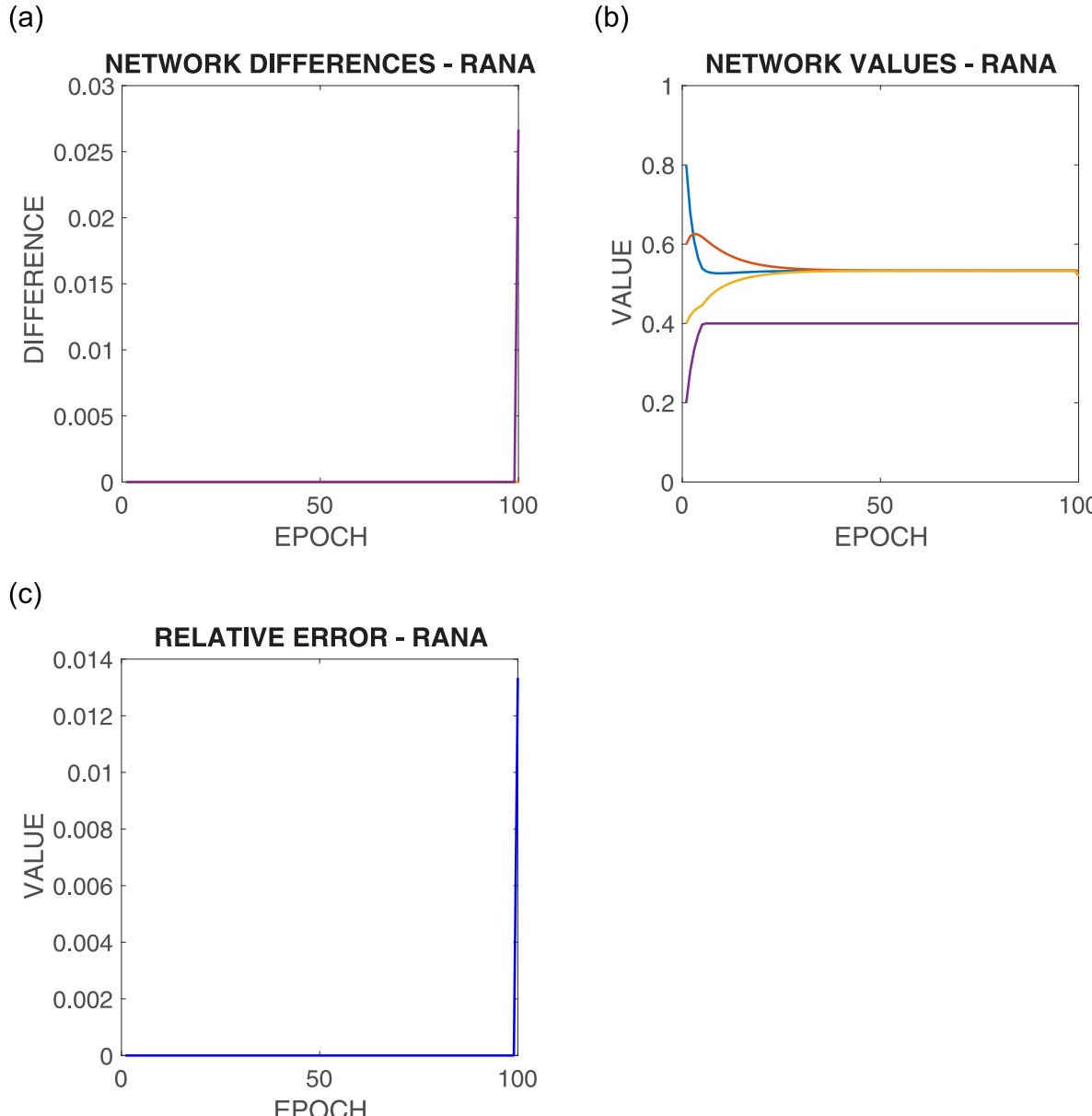

**Fig 7. Supportive consensus process according to the RANA algorithm in the ABCD example.**

So, Fig 8 left shows the evolution of $\delta_i(t)$ in the ABCD example. Fig 8 center shows the evolution of $x_i(t)$ values, and Fig 8 right presents the evolution of $re(t)$.

## Supportive Equity Algorithm (SEA)

The last algorithm presented in this section, Supportive Equity Algorithm (SEA), uses a different approach, based on the renormalization of values. The last variation tries to avoid the double layer and considers the available range directly that each node can assume in case surpluses are detected. Furthermore, the method can manage directly inferior and superior bounds.

(a)

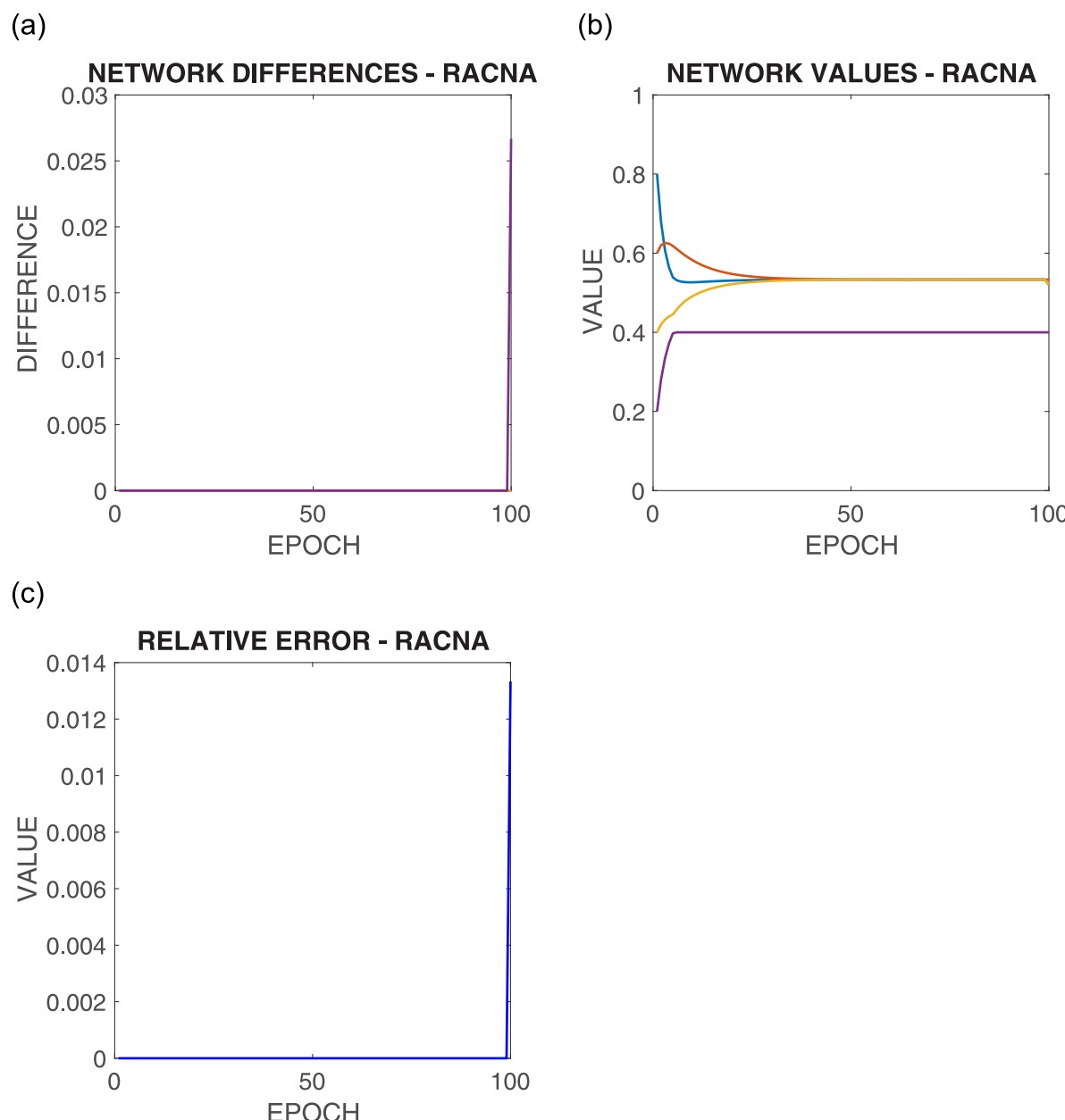

(b)

(c)

**Fig 8. Supportive consensus process according to the RACNA algorithm in the ABCD example.**

We define a change of variable that establishes the proportion of the value of $x_i(t)$ that the node has been moved, taking into account the complete range of movement available.

$$y_i(t) = \frac{x_i(t) - x_i^{INF}}{x_i^{SUP} - x_i^{INF}} = \frac{x_i(t) - x_i^{INF}}{r_i} \qquad (30)$$

and the consensus is performed over the values of $y_i(t)$

$$y_i(t+1) = y_i(t) + \varepsilon \sum_{j \in N_i} [y_j(t) - y_i(t)] \qquad (31)$$

Once the consensus is reached, the change is undone. The final value for each node $x_i(t)$ is the proportional part with which the node contributes. Notice that each node will have a different value, but the sum is conserved.

$$x_i(t+1) = y_i(t+1)r_i + x_i^{INF} \qquad (32)$$

Note that SEA algorithm makes a consensus with transformed variables considering their ranges so that the obtained value represents the proportion of the range each node is going to use. Therefore, no delta is generated since the consensus over the ranges is maintained inside the node bounds.

Fig 9 shows the execution of the SEA algorithm in the *ABCD* example.

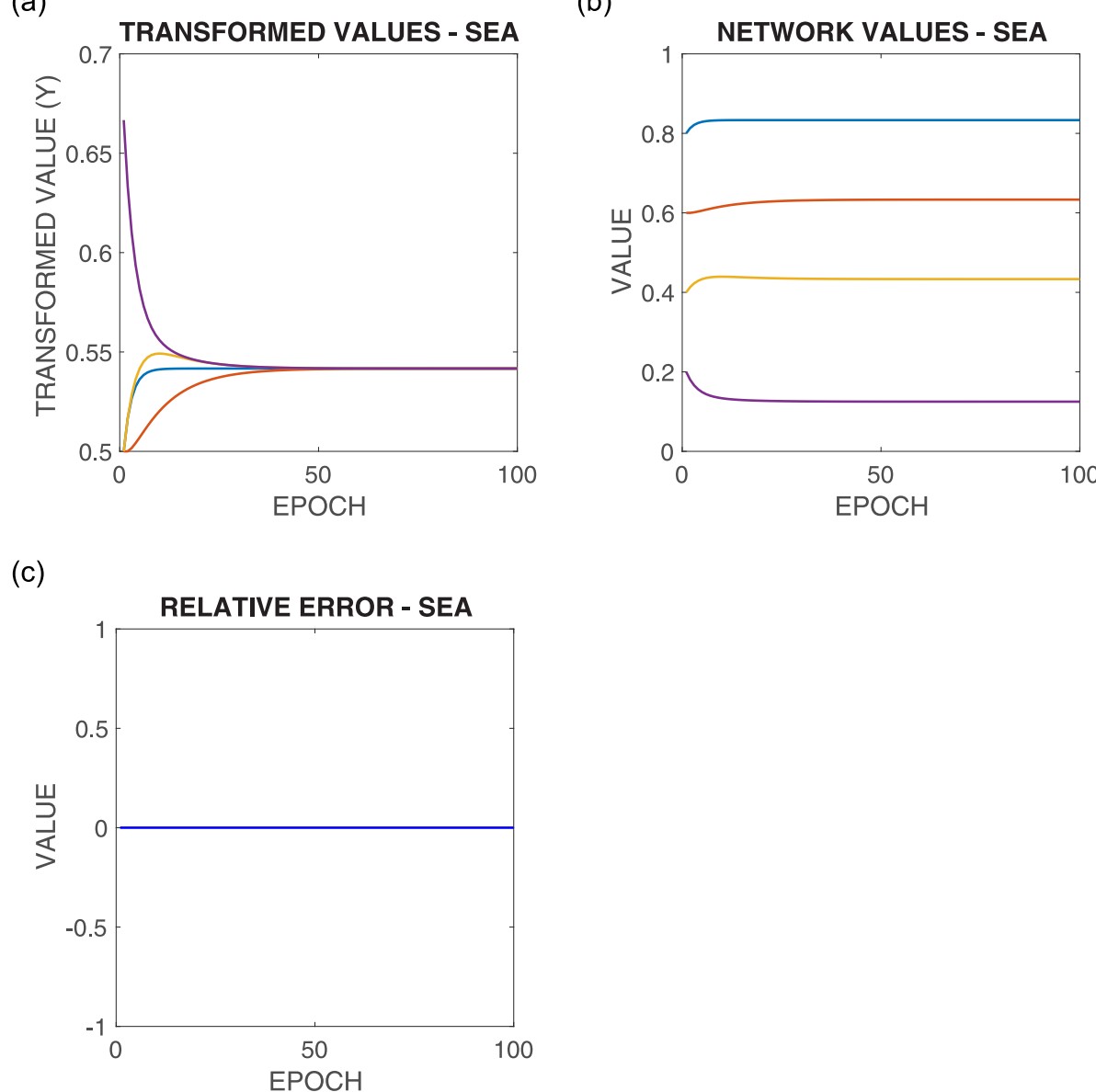

**Fig 9. Supportive consensus process according to the SEA algorithm in the ABCD example.**

So, Fig 9 left shows the evolution of $y_i(t)$ in the ABCD example. Fig 9 center shows the evolution of $x_i(t)$ values, and Fig 9 right presents the evolution of $re(t)$, that is always 0.

### Algorithm summary

The above section has presented two different possible approaches to solve the supportive consensus problem. One of them gives an exact solution, but agents agree on the percentage of their capacity in the solution (the *SEA* algorithm). The other approach is formed by a set of non-exact algorithms (the *CORA*, *ICORA*, *RANA*, and *RACNA algorithms), but agents agree on their value.*

This classification follows two different strategies to solve the problem

1. approximate algorithms, pretending to reach a unique consensus value

2. exact algorithms that reach an agreement maintaining all nodes inside their boundaries

The difference among the first type is how the excess $\delta_i(t)$ is spread throw the network.

It is important to underline that these two different approaches cannot fit the same applications. So, the first one could be applied, for instance, to Smart Grid problems, where agents model energy sources, and the system wants to reach a consensus where all the agents' effort is the same. On the other hand, the second group of algorithms could be applied, for instance, to economic or monetary problems, where a set of agents must agree on the amount of money they must put in a business.

These two approaches are only some possible ones. We do not intend them to be the only ones. Similarly, the algorithms presented are not intended to be the best ones for these approaches, but only some example ones.

## Simulation results

This section presents several experiments done to test the performance of the different supportive consensus algorithms presented above. The first experiment shows the evolution of the network capacity. The second set of experiments test the scalability of the algorithms with different random networks. Finally, the third set study how these algorithms behave with different configurations of the agent's initial boundaries for a random network.

### Network capacities experiment

As Eq 21 proves, the total network's capacity is enough to compensate all the underrange and overrange values outside of the nodes' boundaries. This experiment shows an example of the evolution of the network capacity. It has been tested in an asymmetric random network with 30 nodes. The boundaries have been generated using a Pearson distribution (mean = 0.3, standard deviation = 0.2, and skewness = 1), having the initial value of each agent centered in a range of 0.2 long. This network has been tested with the CORA algorithm, with the results shown in Fig 10. So, Fig 10 left shows the evolution of the network capacities along with the $\Delta(t)$ values during the 500 iterations of the experiment, while Fig 10 right shows the final state of the network showing the final values of each node inside its corresponding range.

Fig 10 left, shows, following the Eq 21, how $\Delta(t)$ values are always between the corresponding $C_{SUP}$ and $C_{INF}$ values. So, in Fig 10 right, the final values of the nodes inside their boundaries can be observed, along with the mean value of the network (represented by a horizontal blue line). Green ranges are over the mean value, red ranges are under the mean value, and blue ranges have the mean value inside the range. It can be observed how green nodes tend to have their value in the lower boundary, while red nodes tend to be in their upper boundary.

(a)

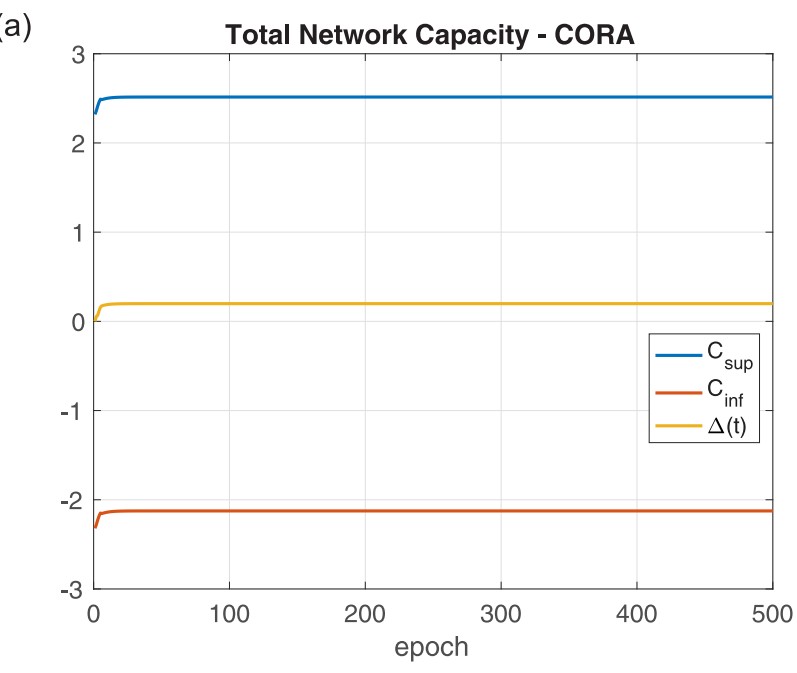

(b)

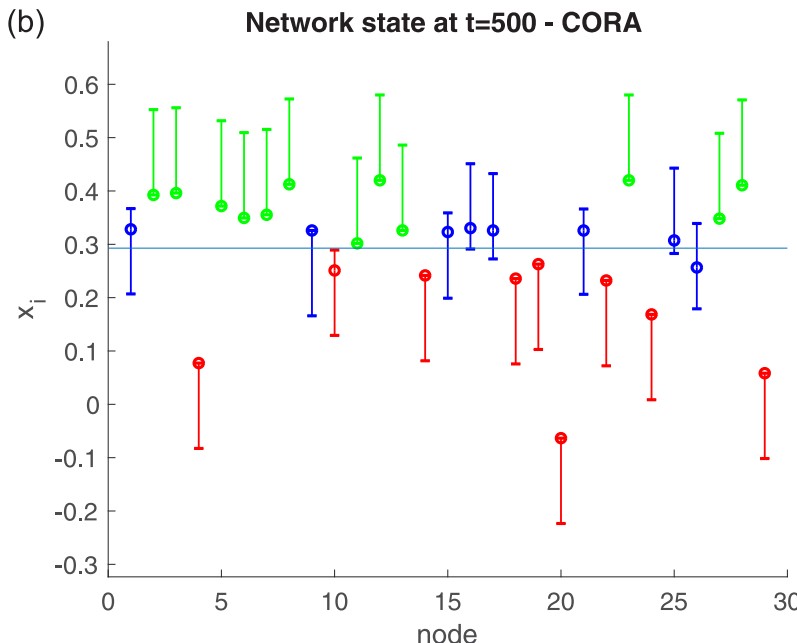

**Fig 10.** (Left) Evolution of the network capacities using the CORA algorithm. (Right) Final values $x_i(t)$ of the nodes.

## Relative error experiment

This experiment compares the results obtained by the different supportive consensus algorithms proposed in the last section with different random networks. The network sizes used are 5, 10, 20, 50, 100, 200, 500, and 1000 nodes, with the density of $\frac{\log n}{n}$ that is the minimum value from which emerges the giant component on random networks. For each network size, there has been generated 500 different networks. Fig 11 shows the evolution of the mean for

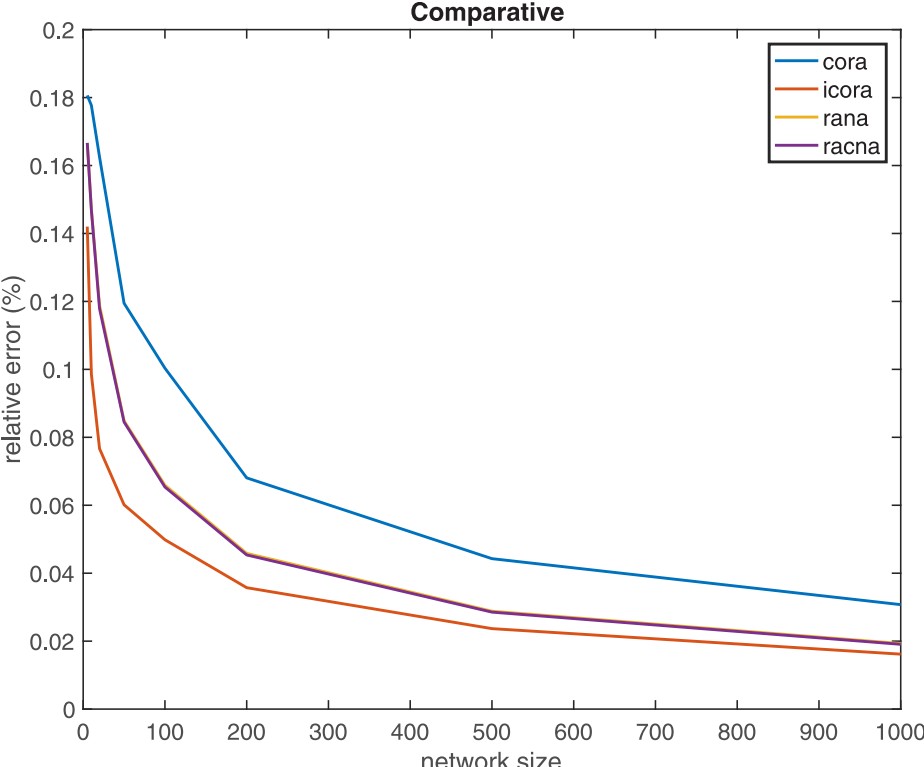

**Fig 11. Relative error experiment: Results of the different supportive consensus proposed using different random networks.**

such 500 networks relative error ($\epsilon(t)$ as declared in Eq 24), each one executed 100 iterations (and 100 more for the internal iterations of the *iCORA* algorithm).

Initial values of nodes have been generated using a uniform distribution in the range [0, 1]. Moreover, each range has been generated so that $x_i^{SUP}$ is $x_i(0)$+ a random value generated in the range [0.2, 0.6], and $x_i^{INF}$ is $x_i(0)$− a random value generated in the range [0.2, 0.6].

Table 1 summarizes the results (mean and standard deviation) obtained for the 500 networks generated for each network size, classified for each one of the different supportive

**Table 1. Relative error experiment: Mean relative error and its corresponding standard deviation.**

| | Algorithm | | | | | | | |
|---|---|---|---|---|---|---|---|---|
| | CORA | | iCORA | | RANA | | RACNA | |
| Nodes | Mean | Dev. | Mean | Dev. | Mean | Dev. | Mean | Dev. |
| 5 | 0,18 | 0,83 | 0,14 | 0,72 | 0,17 | 0,82 | 0,17 | 0,82 |
| 10 | 0,18 | 0,47 | 0,10 | 0,30 | 0,15 | 0,42 | 0,15 | 0,42 |
| 20 | 0,16 | 0,26 | 0,08 | 0,14 | 0,12 | 0,21 | 0,12 | 0,21 |
| 50 | 0,12 | 0,13 | 0,06 | 0,07 | 0,08 | 0,10 | 0,08 | 0,10 |
| 100 | 0,10 | 0,08 | 0,05 | 0,04 | 0,07 | 0,06 | 0,07 | 0,06 |
| 200 | 0,07 | 0,05 | 0,04 | 0,03 | 0,05 | 0,04 | 0,05 | 0,04 |
| 500 | 0,04 | 0,03 | 0,02 | 0,02 | 0,03 | 0,02 | 0,03 | 0,02 |
| 1000 | 0,03 | 0,02 | 0,02 | 0,01 | 0,02 | 0,01 | 0,02 | 0,01 |

consensus algorithms. Remember that, as commented above, no deltas are generated in SEA algorithm and, therefore, the relative error is zero and is not considered in this experiment.

It can be concluded from Fig 11:

- All the presented algorithms work quite well, as their relative error are very small.

- Regarding the relative error, *iCORA* algorithm gives better results.

- As the network size increases, the relative error decreases for all the algorithms.

- All the experiments have very high dispersion, so there cannot be assured a priori that one of these algorithms works better for a particular random network.

### Initial boundaries symmetry experiments

This set of experiments shows how the different proposed supportive consensus algorithms behave with a random network of 50 nodes with 95 links (the average degree of the nodes is $\bar{d} = 3.8$) initial values and boundaries are grouped in different configurations. Each one of these configurations has been executed 500 times over different randomly generated networks, initial values, and boundaries of each node.

**Configuration 1 – $|V^-(0)| > |V^+(0)|$.**   In this experiment, more nodes with boundaries aregreater than the mean of the initial values than nodes with boundaries under the mean. For instance, Fig 12 top left shows one of the 500 random networks generated, where there is the following boundaries distribution: $|V^-(0)| = 26$, $|V^+(0)| = 14$, $|V^*(0)| = 10$. These boundaries have been generated using a Pearson distribution (mean = 0.3, standard deviation = 0.2, and skewness = -1,) having the initial value of each agent centered in a range of 0.1 long.

Fig 13 shows the comparative results of this configuration for the above commented initial situation summarized in Fig 12 top left, where each row corresponds to the results of the execution of one of the different supportive consensus algorithms.

It can be observed from Fig 13:

- Due to the initial asymmetric configuration, most nodes' convergence value is over the mean value.

- As the mean value is outside most nodes boundaries, only a small fraction converges to such value, while the rest is restricted to one of their limits.

- There is no significant difference in the different algorithms' solutions regarding the Relative Error experiment results.

**Configuration 2 – $|V^+(0)| > |V^-(0)|$.**   In this experiment, more nodes exist under the mean of the initial values than nodes with boundaries greater than the mean value. For instance, Fig 12 top right shows one of the 500 random networks generated, where there is the following boundaries distribution: $|V^-(0)| = 14$, $|V^+(0)| = 27$, $|V^*(0)| = 9$. These boundaries have been generated using a Pearson distribution (mean = 0.3, standard deviation = 0.2, and skewness = 1), having the initial value of each agent centered in a range of 0.1 long.

Fig 14 shows the comparative results of this configuration for the above commented initial situation summarized in Fig 12 top right, where each row corresponds to the results of the execution of one of the different supportive consensus algorithms.

It can be observed from Fig 14:

- Due to the initial asymmetric configuration, most nodes' convergence value is under the mean value.

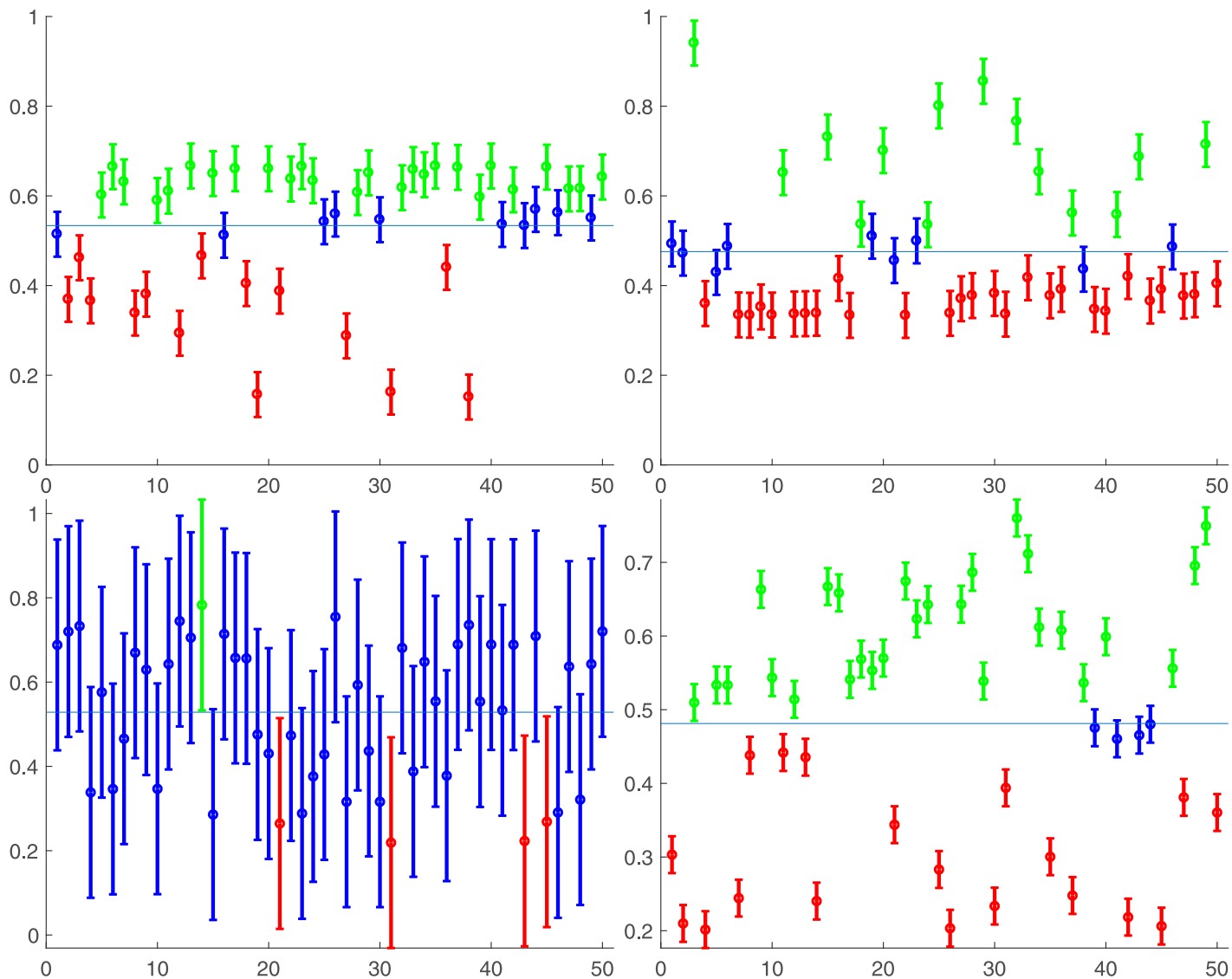

**Fig 12. One particular example of each configuration of the experiment.** Four initial configurations are considered: two asymmetric (top), where are more nodes over or below the average value, and symmetric (bottom), where the proportion of nodes that over or underrange is the same.

- Like in the previous configuration, as the mean value is outside most node boundaries, only a small value converges to such value, while the rest is restricted to one of their limits.

- There is no significant difference in the different algorithms' solutions regarding the Relative Error experiment results.

**Configuration 3 − $|V^*(0)| \gg |V^+(0)| + |V^-(0)|$.**   In this experiment, most of the nodes have the mean of the initial values inside their boundaries. For instance, Fig 12 bottom left shows one of the 500 random networks generated, where there is the following boundaries distribution: $|V^-(0)| = 1$, $|V^+(0)| = 4$, $|V^*(0)| = 45$. These boundaries have been generated using a uniform distribution with each agent's initial value centered in a range of 0.5 long.

Fig 15 shows the comparative results of this configuration for the above commented initial situation summarized in Fig 12 bottom left, where each row corresponds to the results of the execution of one of the different supportive consensus algorithms.

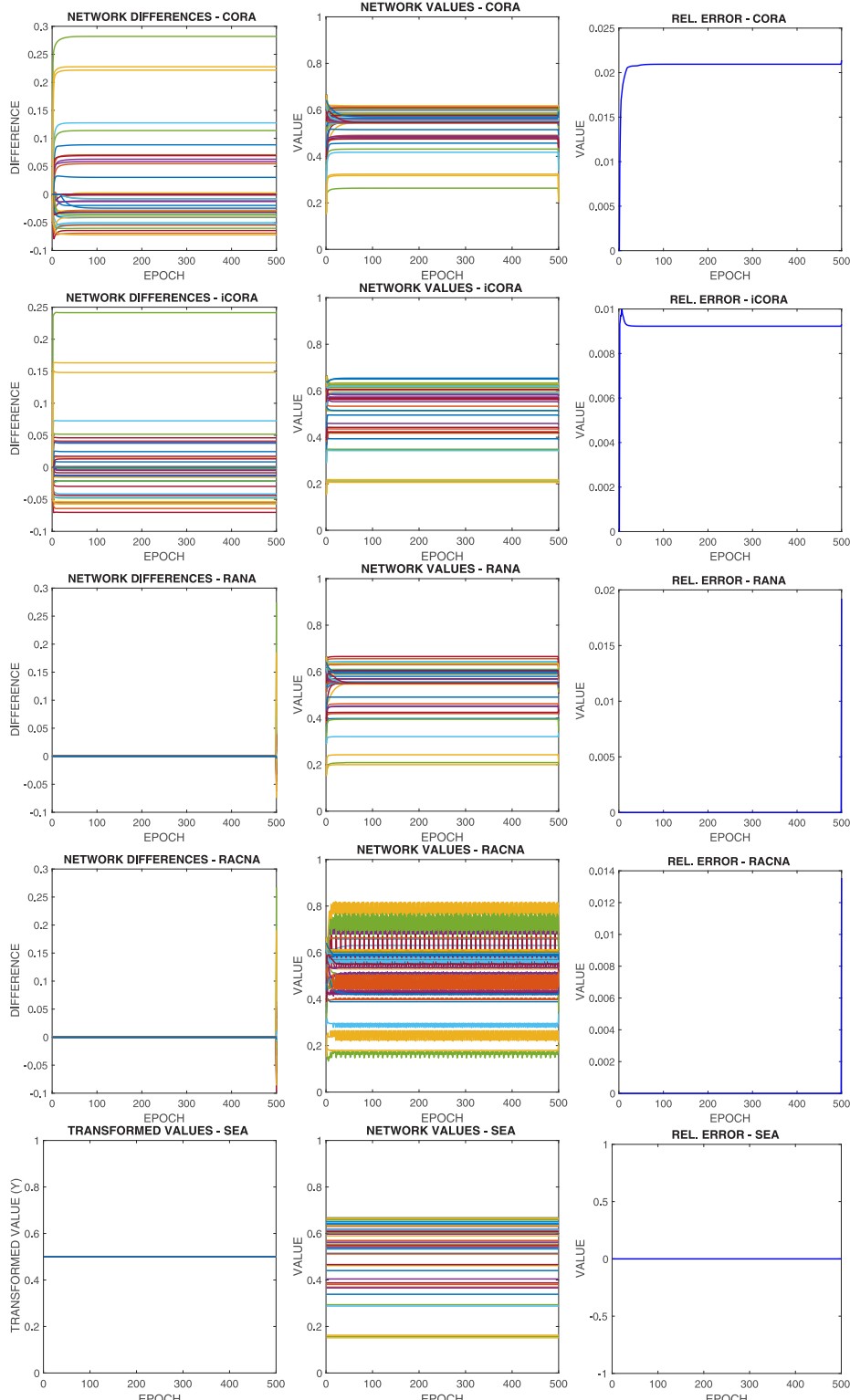

**Fig 13. Results of configuration 1 with initial values as in Fig 12 top left, where there are more nodes whose lower bound is over the average value.** Each row shows one of the algorithms $|V^-(0)| > |V^+(0)|$.

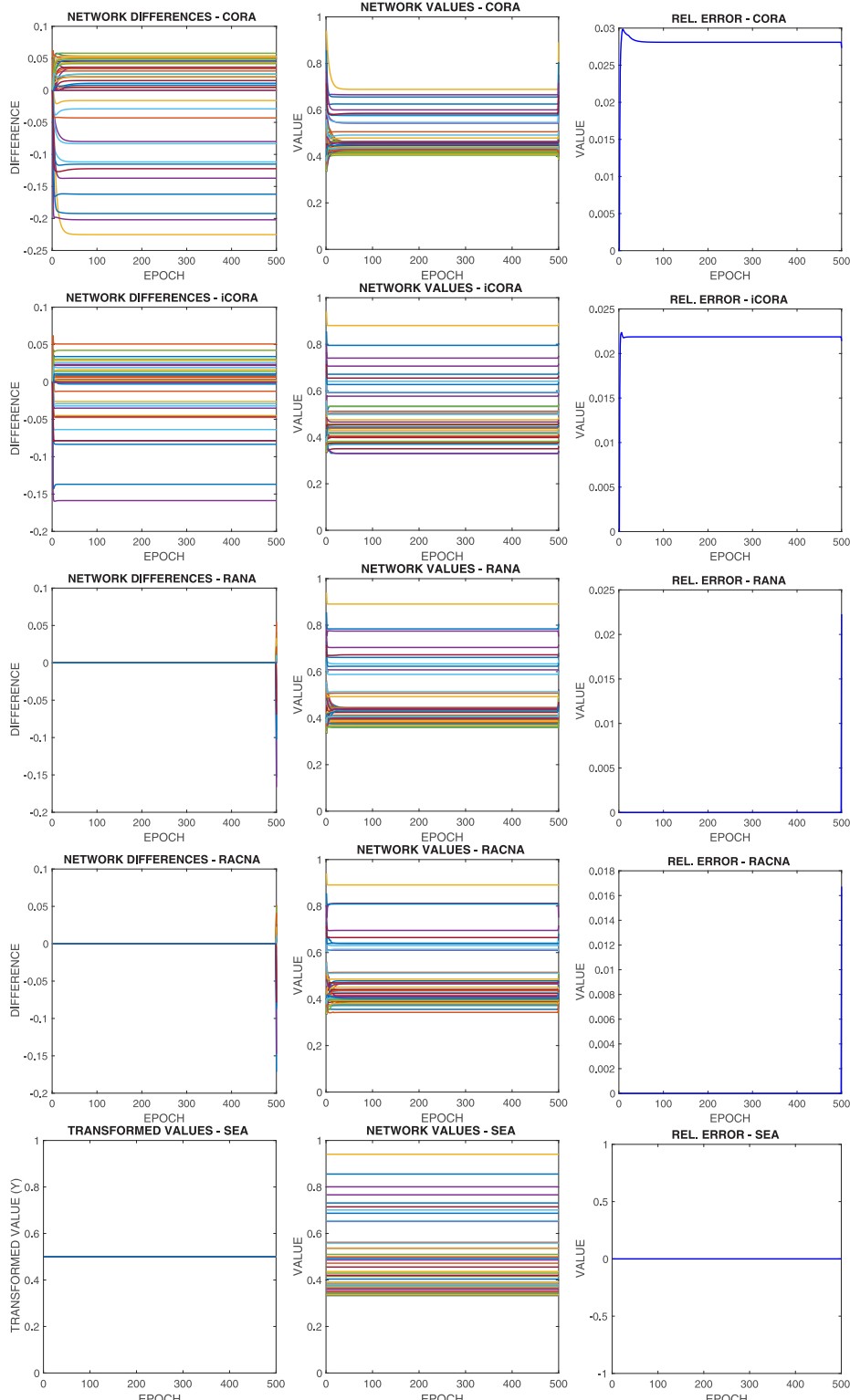

**Fig 14. Results of configuration 2 with initial values as in Fig 12 top right, where there are more nodes whose upper bound is under the average value $|V^+(0)| > |V^-(0)|$.** Each row shows one of the algorithms.

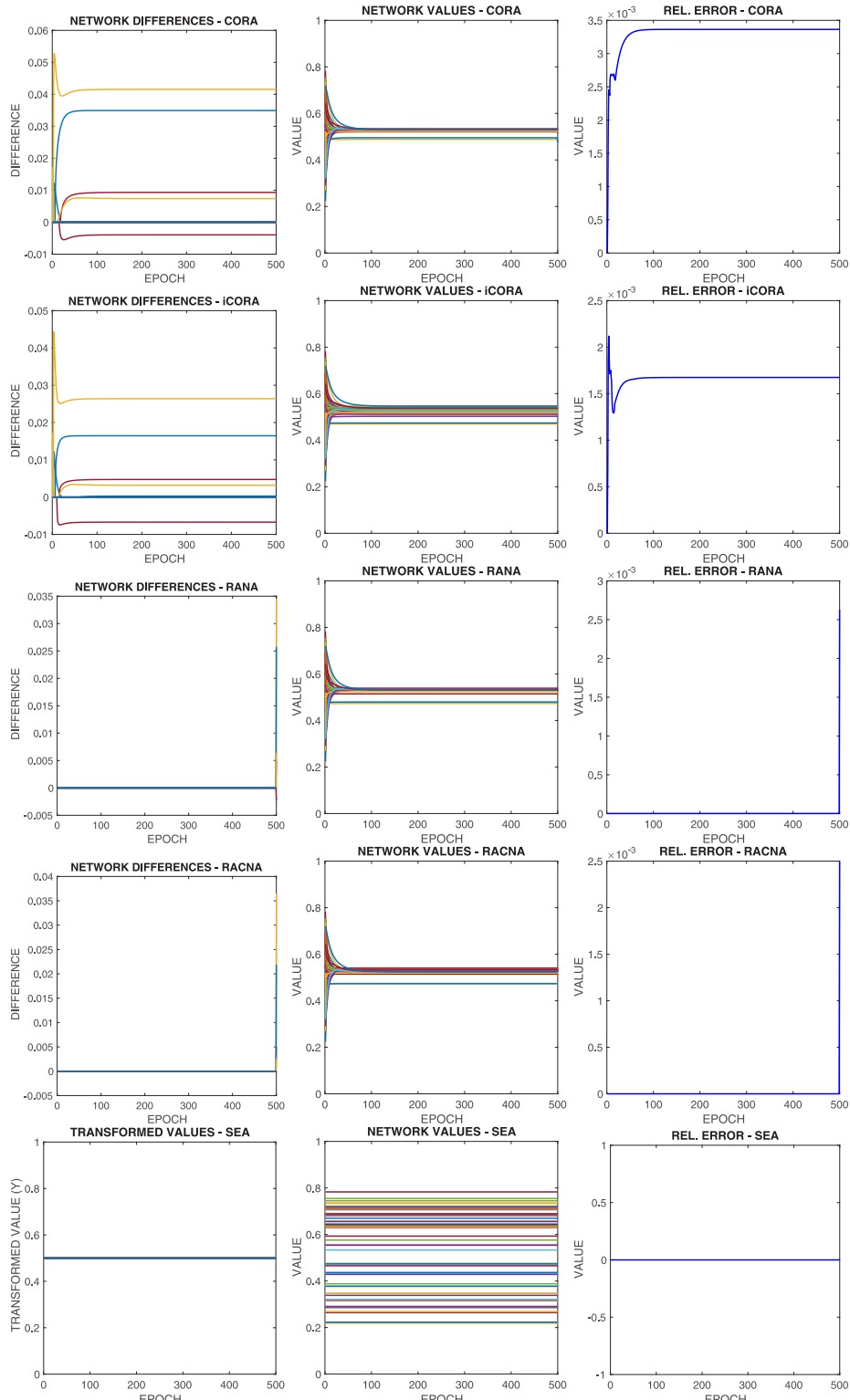

**Fig 15. Results of configuration 3 with initial values as in Fig 12 bottom left, where there the average value is into the bounds of the majority of the nodes $|V^*(0)| >> |V^+(0)| + |V^-(0)|$.** Each row shows one of the algorithms.

It can be observed from Fig 15:

- As most of the nodes have the mean value inside their boundaries, their convergence values are very close to the compensated mean value, that is, the value that compensates the nodes with the mean value is outside their boundaries ($V^-(t) \cup V^+(t)$).

- Once again, there is no significant difference in the different algorithms' result regarding the results shown in the Relative Error experiment.

**Configuration 4 – $|V^*(0)| \rightarrow 0$.**   In this experiment, almost no node includes the mean of the initial values inside its boundaries. For instance, Fig 12 bottom right shows one of the 500 random networks generated, where there is the following boundaries distribution: $|V^-(0)| = 27, |V^+(0)| = 19, |V^*(0)| = 4$. These boundaries have been generated using a uniform distribution with each agent's initial value of each agent centered in a range of 0.1 long.

Fig 16 shows the comparative results of this configuration for the above commented initial situation summarized in Fig 12 bottom right, where each row corresponds to the results of the execution of one of the different supportive consensus algorithms.

It can be observed from Fig 16:

- As most nodes have the mean value outside their boundaries, they converge to one of their limits. Nevertheless, as Eq 21 claims, the network convergence is assured, and so, the nodes would compensate between them for the mean value.

- Once again, there is no significant difference in the different algorithms' result regarding the results shown in the Relative Error experiment.

**Configurations comparison.**   Table 2 shows the results of the 500 repetitions of each one of the presented configurations. It has to be taken into account that *SEA* algorithm has not been considered in these configurations because of the way it is implemented because it does not have any relative error to be considered. It is important to remark that as commented in Eq 21, it does not matter if the initial configuration of the network is asymmetric or not: the supportive consensus will converge.

The asymmetric configurations give more relative error than the symmetric ones because there are few agents with the capacity to compensate out-of-boundaries agents. The relative error obtained by the proposed algorithms in these experiments is comparatively coherent to the Relative Error experiment results.

According to the obtained results shown in Table 2, independently of the symmetric or initial asymmetric configuration of the network, *CORA* algorithm obtains the worst results regarding Relative Error, while *iCORA* algorithm gets the best ones.

It has to be underlined the results obtained by the different algorithms in Configuration 4. This configuration is an extreme situation for traditional consensus because almost all the nodes have the network mean value outside their boundaries. As commented above, the traditional consensus is cannot to give a solution for this situation, while a *supportive consensus* can give a satisfying solution for all the participating nodes. Moreover, Table 2 shows that even in this extreme situation, the proposed algorithms give an acceptable Relative Error for real situations where this *supportive consensus* is required.

## Conclusions

This paper presents an innovative kind of consensus where all the participants may have their acceptable values bounded, and the classical consensus mean value is outside of some agents'

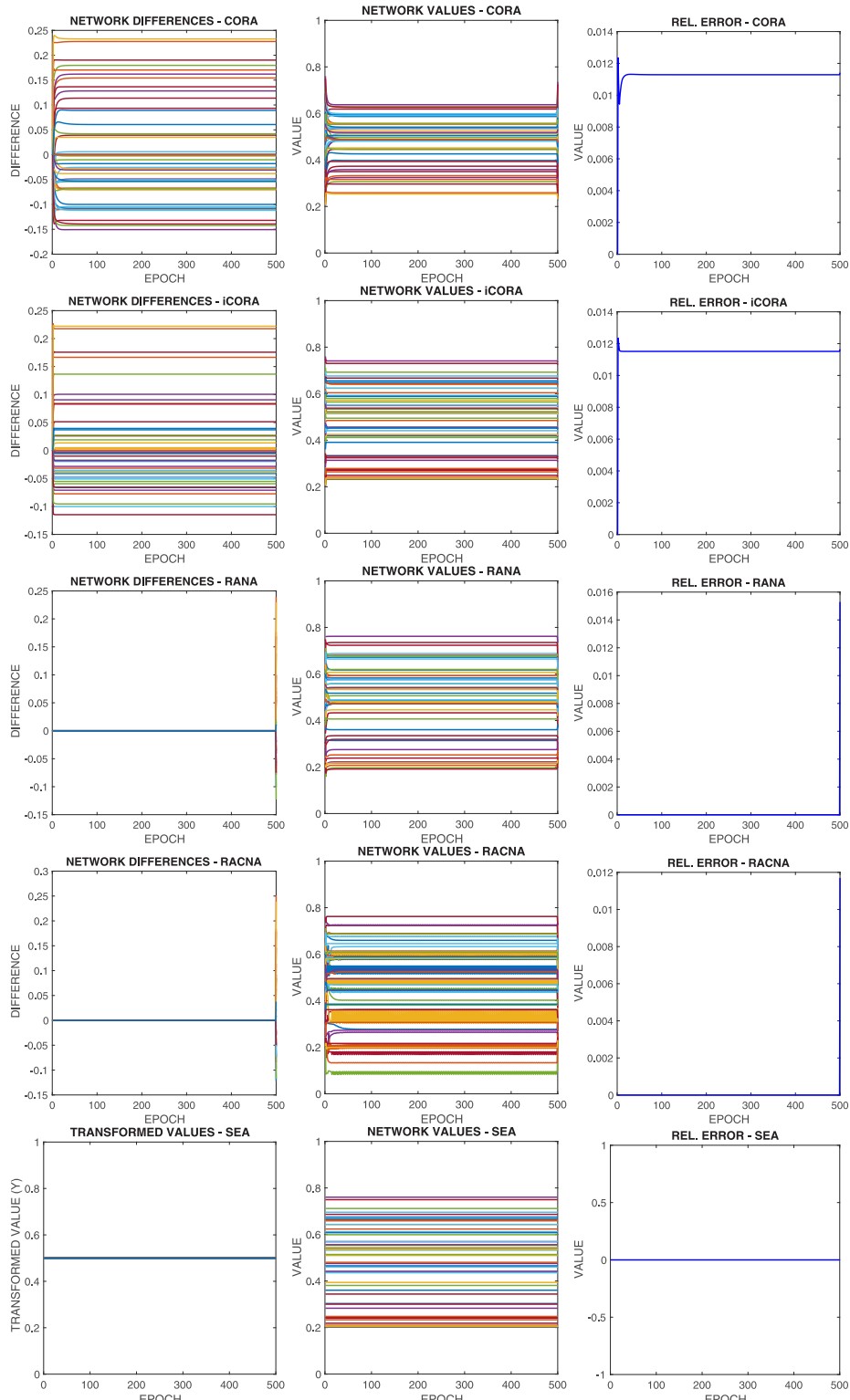

**Fig 16. Results of configuration 4 with initial values as in Fig 12 bottom right, where the solution is out of the bounds of almost all the nodes $|V^*(0)| \to 0$.** Each row shows one of the algorithms.

**Table 2. Mean and standard deviation of the relative error of the 500 experiments for each configuration.**

| Set | Algorithm | | | | | | | |
|---|---|---|---|---|---|---|---|---|
| | CORA | | iCORA | | RANA | | RACNA | |
| | Mean | Dev. | Mean | Dev. | Mean | Dev. | Mean | Dev. |
| Conf. 1 | 3.1265 | 1.0125 | 1.9165 | 0.8317 | 2.7001 | 1.0104 | 2.1755 | 1.1217 |
| Conf. 2 | 3.3262 | 1.3180 | 2.0671 | 0.9491 | 2.9057 | 1.2281 | 2.3677 | 1.3947 |
| Conf. 3 | 1.1098 | 0.8766 | 0.6429 | 0.5333 | 0.8452 | 0.6910 | 0.8348 | 0.6867 |
| Conf. 4 | 0.8771 | 0.6638 | 0.7522 | 0.5766 | 0.8543 | 0.6443 | 1.1756 | 1.3409 |

boundaries. Such classical consensus is unable to solve this problem with all agents participating in such a solution.

Moreover, there are some equality solutions to this problem in the literature of constrained consensus, while we present an equity approach. This proposal tries to compensate for the lack of such agents by being assumed for other agents. The paper shows how the system has capacity enough to compensate for the lack of agents reaching their limits if all the involved agents have their initial value inside their corresponding boundaries. Different algorithms are proposed to implement different approaches to this supportive consensus regarding how this compensation is made, including equality deliveries and equity ones. We have presented several different simulation experiments to show the different proposed algorithms' performance, obtaining satisfying solutions with all of them.

The supportive consensus is a new problem field with many applications to real-life problems that are open to new algorithms and proposals.

## Author Contributions

**Formal analysis:** A. Palomares, M. Rebollo, C. Carrascosa.

**Software:** A. Palomares, M. Rebollo, C. Carrascosa.

**Visualization:** A. Palomares, M. Rebollo, C. Carrascosa.

**Writing – original draft:** A. Palomares, M. Rebollo, C. Carrascosa.

**Writing – review & editing:** A. Palomares, M. Rebollo, C. Carrascosa.

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
