## [Decision Letter · Decision Letter 0]

24 Jul 2020

PONE-D-20-21157

Supportive Consensus

PLOS ONE

Dear Dr. Palomares Chust,

Thank you for submitting your manuscript to PLOS ONE. After careful consideration, we feel that it has merit but does not fully meet PLOS ONE’s publication criteria as it currently stands. Therefore, we invite you to submit a revised version of the manuscript that addresses the points raised during the review process.

The associate editor received two consistent reviews. Both reviewers suggested that a major revision of the manuscript is needed before it can be accepted for publication. They also pointed out that the paper contains numerous grammatical mistakes and typos. Reviewer 1 suggested that the contribution of this paper should be highlighted. S/He also pointed out that the assumptions should be stated explicitly, and the convergence rate should be established. Reviewer 2 raised very specific comments regarding the presentation of the paper. Overall, I think this paper would be acceptable for publication if the above points are addressed convincingly, which would require a substantial revision. I therefore recommend major revision and re-review for this manuscript.

We look forward to receiving your revised manuscript.

Kind regards,

Fei Chen

Academic Editor

PLOS ONE

Journal Requirements:

2. Please upload a copy of Figure ??, to which you refer in your text on page 16. If the figure is no longer to be included as part of the submission please remove all reference to it within the text.

Additional Editor Comments (if provided):

The associate editor received two consistent reviews. Both reviewers suggested that a major revision of the manuscript is needed before it can be accepted for publication. They also pointed out that the paper contains numerous grammatical mistakes and typos. Reviewer 1 suggested that the contribution of this paper should be highlighted. S/He also pointed out that the assumptions should be stated explicitly, and the convergence rate should be established. Reviewer 2 raised very specific comments regarding the presentation of the paper. Overall, I think this paper would be acceptable for publication if the above points are addressed convincingly, which would require a substantial revision. I therefore recommend major revision and re-review for this manuscript.

Reviewers' comments:

Reviewer's Responses to Questions

**Comments to the Author**

1. Is the manuscript technically sound, and do the data support the conclusions?

Reviewer #1: Yes

Reviewer #2: Yes

2. Has the statistical analysis been performed appropriately and rigorously? 

Reviewer #1: No

Reviewer #2: Yes

3. Have the authors made all data underlying the findings in their manuscript fully available?

Reviewer #1: Yes

Reviewer #2: Yes

4. Is the manuscript presented in an intelligible fashion and written in standard English?

Reviewer #1: Yes

Reviewer #2: Yes

5. Review Comments to the Author

Reviewer #1: This paper proposes multiple supportive consensus algorithms for multi-agent systems where each agent has a set constraint. Especially, the authors consider the case that there is no intersection among the set constraints of all the agents. The topic is interesting. However, it seems that this paper is not well-written. My comments are as follows.

1) There are many grammatical mistakes in this paper. For example, in page 1 line 2, it should be “to deal with” instead of “to dealing with”. In page 1 line 14, it should be “must obtain” instead of “must to obtain”. Please check the whole paper carefully to correct all the grammatical mistakes.

2) In page 3 line 80, the authors state the contributions of this paper in this paragraph. However, the introduction of the contributions is a little pale. I suggest presenting a more detailed introduction to the contributions here. For example, what algorithms do you propose? What are the differences between these proposed algorithms? How do you analyze the algorithms? How do you verify the effectiveness of the proposed algorithms?

3) The assumptions of the proposed algorithms should be stated clearly. For example, can the proposed algorithm work under directed graphs? Is the underlying graph required to be connected? Can the proposed algorithms work under arbitrary initial conditions?

4) The goals of these proposed algorithms should be stated clearly. Why do you need multiple algorithms to solve the same problem? What are the pros and cons of each algorithm?

5) Convergence properties of the proposed algorithms should be stated clearly. For example, what is the convergence rate? What are the exact convergent values?

Reviewer #2: This paper presents an innovative kind of consensus, named as the Supportive Consensus. All the agents considered in this work may have their acceptable values bounded, and the classical consensus mean value is outside of the boundaries of some agents. For such case, the classical consensus is unable to give a solution while the Supportive Consensus protocol can. The idea of this work is novel, while the composition of the paper is terrible.

(1) It is suggested that the body of each graph appear with its corresponding title such Figure 4, 5, and the others.

(2) It is suggested that we call them table rather than figure for the Figures 1,2,3 in the current manuscript.

(3) The definition of degree, which is used in line 117, is suggested to be provided in this work. Moreover, it is suggested that both the in-degree and the out-degree are provided.

(4) The section number is left in line 179, “, as defined in Section .”.

(5) The usage of ${{x}_{i}}(t)$ is not appropriate in lines 182, 184 and 186. As we know, ${{x}_{i}}(t)$ is the state of node $i$, and it is not the consensus value.

(6) There are many redundant “=” in the equation below line 285.

(7) The English should be improved since there are many grammatical problems such as “This paper presents a novel approach to dealing with…”(line 2), “, and we proof that if …”(line 22). There is no need to indent the beginning of some paragraphs such as the paragraphs on line 119 and line 124.

6. PLOS authors have the option to publish the peer review history of their article (what does this mean?). If published, this will include your full peer review and any attached files.

Reviewer #1: No

Reviewer #2: No

---

## [Author Response · Author response to Decision Letter 0]

9 Nov 2020

November 6, 2020 

Fei Chen 

Academic Editor 

PLOS ONE 

Subject: Submission of revised paper PONE-D-20-21157, ’Supportive Consensus’. 

Dear Dr. Chen, 

Thank you for your e-mail dated 24th July 2020, enclosing the reviewers’ com­ments. We have carefully reviewed the comments and have revised the manuscript accordingly. Our responses are given in a point-by-point manner below. Changes to the document are shown in blue. 

We have rechecked the language, and the mistakes detected in the reading proof corrected. The paper’s contribution has been improved in a new section that explains the main results and the di.erences between the approaches we propose. Regarding the convergence analysis, we have included a new section that demonstrates the solution in a scenario with perfect information regarding the solution obtained by the algorithms. The convergence of those algorithms is ensured by the original work of Olfati–Saber and Murray, in which we base our proposal. As the algorithms match the requirements, stability is ensured. We have highlighted this fact in the paper too. The rest of the observations have been appropriately addressed, in our opinion. 

We hope the revised version is now suitable for publication and look forward to hearing from you in due course. 

Sincerely, 

Dr. Alberto Palomares . 

Associate Professor 

Universitat Politecnica de Valencia (Spain) 

*Response to reviewer 1 

Thank you for your review of our paper. We have answered each of your points below 

1. There are many grammatical mistakes in this paper. For example, in page 1 line 2, it should be ”to deal with” instead of ”to dealing with”. In page 1 line 14, it should be ”must obtain” instead of ”must to obtain”. Please check the whole paper carefully to correct all the grammatical mistakes. 

We have rechecked the language. The mistakes indicated by the reviewer have been corrected, and some others detected in the reading proof. 

2. In page 3 line 80, the authors state the contributions of this paper in this paragraph. However, the introduction of the contributions is a little pale. I suggest presenting a more detailed introduction to the contributions here. For example, what algorithms do you propose? What are the differences between these proposed algorithms? How do you analyze the algorithms? How do you verify the effectiveness of the proposed algorithms? 

We have included in the Scope section a broader explanation detailing the information required by the reviewer. In this part, we have described the algorithm, explain their differences in the approach to the supportive consensus problem, and the experiments to evaluate their performance and the quality of the solutions they reach. 

3. The assumptions of the proposed algorithms should be stated clearly. For example, can the proposed algorithm work under directed graphs? Is the underlying graph required to be connected? Can the proposed algorithms work under arbitrary initial conditions? 

We have clarified these characteristics of the network in its definition. We assume that the consensus process runs over a non-directed, strongly connected network. Algorithms could be extended to the directed case but keeping the connection condition. ”Let G = {V, E} an undirected, strong connected graph with n nodes, V = {1, ..., n}, and e edges E . V × V , where (i, j) . E if there exists a link between nodes i and j.” Regarding the initial condition, the only consideration is that all the nodes must begin inside their ranges (see explanation after Eq. 21) We have remarked this fact. ”In general, assuming that the initial values of all agents (t=0) are within their bounds, the network globally always (t) has the capacity to compen­sate the total amount of values[. . . ]” 

4. The goals of these proposed algorithms should be stated clearly. Why do you need multiple algorithms to solve the same problem? What are the pros and cons of each algorithm? 

We have added a paragraph at the beginning of the section ’Supportive Consensus Algorithms Proposal’ explaining the goals and the reasons for the different algorithms. 

2 

’The algorithm that we have called SEA[. . . ]’ 

Furthermore, we have included a section ’Algorithm Summary’ summa­rizing the pros and cons of the proposed algorithms. 

5. Convergence properties of the proposed algorithms should be stated clearly. For example, what is the convergence rate? What are the exact convergent values? 

We have included a new section that calculates the solution in a sce­nario with perfect information regarding the solution obtained by the al­gorithms. We have considered the centralized solution as the exact solu­tion. The results obtained by the algorithms are approximations of this value. The convergence of the proposed algorithms is ensured by the original work of Olfati–Saber and Murray in which we base our proposal. As the algorithms match the requirements, stability is ensured. We have high­lighted this fact in the paper, too, related to the sum conservation. ’As Algorithm conserves the sum of the initial values and fulfil the condi­tions of the Olfati–Saber consensus algorithm to converge,(Footnote: The convergence ...) we need to ensure that the Supportive Consensus func­tion also conserves the sum. We check this condition in each one of the proposed algorithms.’ 

*Response to reviewer 2 

Thank you for your comments. Our answers to your points are as follow 

1. It is suggested that the body of each graph appear with its corresponding title such Figure 4, 5, and the others. 

Following the submission rules of PLOS, figures have to be moved to the end of the paper and keep the captions in place. 

2. It is suggested that we call them table rather than figure for the Figures 1,2,3 in the current manuscript. 

These figures contain both the table and the .figure, and as in the previous consideration. They appear at the end of the document. 

3. 

The definition of degree, which is used in line 117, is suggested to be provided in this work. Moreover, it is suggested that both the in-degree and the out-degree are provided. We have included the definition of the degree of the network ” In graph theory, the degree of a node di is the number of edges that are incident to the node, and therefore |Ni| = di.”. As we assume a non-directed network, the definition of in-and out-degree are not necessary. 

4. 

The section number is left in line 179, ”, as defined in Section .”. We have included the name of the section in the reference (Background) 

5. 

The usage of xi(t) is not appropriate in lines 182, 184 and 186. As we know, xi(t) is the state of node i, and it is not the consensus value. 

We do not make any difference between the state of the node and its value and we use both as synonyms. The value xi(t) is the only information stored in the nodes. Its value is defined and bounded in equation 5. 

6. There are many redundant ”=” in the equation below line 285. We have removed the equal signs at the end of each line of equation 5 

7. The English should be improved since there are many grammatical prob­lems such as ”This paper presents a novel approach to dealing with...” (line 2), ”, and we proof that if ...” (line 22). There is no need to indent the beginning of some paragraphs such as the paragraphs on line 119 and line 124. We have rechecked the language. The mistakes indicated by the reviewer have been corrected, and some others detected in the reading proof.

---

## [Decision Letter · Decision Letter 1]

18 Nov 2020

Supportive Consensus

PONE-D-20-21157R1

Dear Dr. Palomares Chust,

We’re pleased to inform you that your manuscript has been judged scientifically suitable for publication and will be formally accepted for publication once it meets all outstanding technical requirements.

Kind regards,

Fei Chen

Academic Editor

PLOS ONE

Additional Editor Comments (optional):

The revised paper has been evaluated by the same reviewers. They both agreed that the paper is publishable, but also raised some minor comments. Please address them in the final version. Based on their comments, I am pleased to recommend it for publication.

Reviewers' comments:

Reviewer's Responses to Questions

**Comments to the Author**

1. If the authors have adequately addressed your comments raised in a previous round of review and you feel that this manuscript is now acceptable for publication, you may indicate that here to bypass the “Comments to the Author” section, enter your conflict of interest statement in the “Confidential to Editor” section, and submit your "Accept" recommendation.

Reviewer #1: All comments have been addressed

Reviewer #2: (No Response)

2. Is the manuscript technically sound, and do the data support the conclusions?

Reviewer #1: Yes

Reviewer #2: Yes

3. Has the statistical analysis been performed appropriately and rigorously? 

Reviewer #1: Yes

Reviewer #2: Yes

4. Have the authors made all data underlying the findings in their manuscript fully available?

Reviewer #1: Yes

Reviewer #2: Yes

5. Is the manuscript presented in an intelligible fashion and written in standard English?

Reviewer #1: Yes

Reviewer #2: Yes

6. Review Comments to the Author

Reviewer #1: The authors have addressed all the comments that the reviewer raised. The reviewer's only concern is presented below.

1)In the background section, the authors stated "Let G = {V, E} an undirected, strong connected graph.....". However, it seems that the definition of "strongly connected" is only for directed graph. Please double check with the definition of "strongly connected".

Reviewer #2: A new kind of algorithms called supportive consensus is presented in this work. Most of the suggestions from the reviewers have been settled in this round.

(1) In the example for Algorithm 1, it is given that $R=\\{[6,10],[4,6],[2,5.2],[0,4]\\}$, which means that ${{x}_{A}}\\in [x_{A}^{INF},x_{A}^{SUP}]=[6,10]$. The solution of A is $x_A=5.4$, which is outside the scope of $[6,10]$. I think the authors should check the Algorithm 1 and its examples.

(2) The title of Table 2 should be placed at the top of the table.

(3) There are still some grammatical problems remained in this version of the manuscript. For example, “Let $G=\\{V,E\\}$ an undirected, strong connected graph” (line 134) should be “Let $G=\\{V,E\\}$ be an undirected, strong connected graph”. In line 204, “Lets call” should be “Let’s call”.

7. PLOS authors have the option to publish the peer review history of their article (what does this mean?). If published, this will include your full peer review and any attached files.

Reviewer #1: No

Reviewer #2: No

---

## [Editor Report · Acceptance letter]

1 Dec 2020

PONE-D-20-21157R1 

Supportive Consensus 

Dear Dr. Palomares:

I'm pleased to inform you that your manuscript has been deemed suitable for publication in PLOS ONE. Congratulations! Your manuscript is now with our production department. 

Kind regards, 

on behalf of

Dr. Fei Chen 

Academic Editor

PLOS ONE